# Specialisation of meiotic kinetochores revealed through a synthetic spindle assembly checkpoint strategy

Lori B Koch[1], Tiasha Ghosh[1], Christos Spanos[1,2], Adèle L Marston[1,2]*

[1]Centre for Cell Biology, School of Biological Sciences, University of Edinburgh, Edinburgh, United Kingdom; [2]Discovery Research Platform for Hidden Cell Biology, School of Biological Sciences, University of Edinburgh, Edinburgh, United Kingdom

## eLife Assessment

Koch et al. describe a **valuable** novel methodology, SynSAC, to synchronise cells to analyse meiosis I or meiosis II or mitotic metaphase in budding yeast. The authors present **convincing** data to validate abscisic acid-induced dimerisation to induce a synthetic spindle assembly checkpoint (SAC) arrest that will be of particular importance to analyse meiosis II. The authors use their approach to determine the composition and phosphorylation of kinetochores from meiotic metaphase I and metaphase II that will be of interest to the broader meiosis research community.
[Editors' note: this paper was reviewed by Review Commons.]

*For correspondence:
adele.marston@ed.ac.uk

Competing interest: The authors declare that no competing interests exist.

**Abstract** Meiosis creates haploid gametes through two sequential M phases. While many studies have focused on meiosis I, the molecular events which drive and define meiosis II are largely unknown. Here, we report a novel cell synchronisation strategy which allows for collection of budding yeast *Saccharomyces cerevisiae* cells arrested at metaphase I or metaphase II, enabling better characterisation of meiosis II events. The method relies on chemically-inducible dimerisation of ectopic copies of spindle assembly checkpoint (SAC) proteins Mps1 and Spc105. Using this synthetic SAC (SynSAC) approach, we found that the SAC response is weaker in metaphase I compared to metaphase II and that the PP1 binding site within Spc105 contributes to restraining the MI SAC response. Furthermore, we demonstrate the utility of the SynSAC approach by analysing the composition and phosphorylation of kinetochores from metaphase I and metaphase II. This revealed an increase in the abundance of outer kinetochore proteins in meiotic metaphase I and reduced phosphorylation on metaphase II kinetochore proteins. Overall, we present the SynSAC method as a valuable tool for analysis of both meiotic metaphases.

## Introduction

To make haploid gametes for sexual reproduction, diploid cells undergo meiosis which consists of a single round of DNA replication followed by two sequential chromosome segregation events, meiosis I and meiosis II. In meiosis I, maternal and paternal chromosomes, called homologs, separate, while in meiosis II, sister chromatids separate, similar to mitosis. To achieve this, the sister chromatids within each homolog are mono-oriented toward the same spindle pole in meiosis I but bi-oriented towards opposite poles in meiosis II. This requires that kinetochores, large protein complexes which connect chromosomes to spindle microtubules, establish mono-orientation in meiosis I and then convert to bi-orientation in meiosis II. The sequential segregation of homologs and sister chromatids further rely on the step-wise loss of sister chromatid cohesion coupled to two consecutive rounds of spindle

assembly and disassembly (reviewed in *Duro and Marston, 2015*). How these key cell biological events are established and coupled in meiosis to allow two distinct rounds of chromosome segregation without an intervening S phase is not well understood. In human oocytes, meiotic chromosome segregation is error-prone and aneuploidy is common (*Gruhn and Hoffmann, 2022*). Thus, identifying molecular mechanisms which prevent improper chromosome segregation may help efforts to improve fertility.

A major barrier in defining molecular events that direct the two meiotic divisions is a lack of synchronisation tools which allow collection of populations of cells at key meiotic stages. While mammalian oocytes naturally arrest in metaphase II, material is limited, which precludes biochemical analysis. *Xenopus* oocytes provide a powerful system for analysis of metaphase II extracts, but isolation of pure metaphase I populations is challenging (*Iwabuchi et al., 2000*; *Peshkin et al., 2025*). Furthermore, identifying chromosome-related proteins in the comparatively large cytoplasmic volume of oocytes presents an additional impediment.

The ability to harvest large number of cells together with ease of genetic manipulation have made yeast outstanding tools for discovering the fundamental mechanisms of meiosis. However, compared to mitosis, our knowledge of meiotic chromosome segregation has lagged behind, in part due to a lack of robust synchronisation systems. Nevertheless, recent advances allow reversible arrest of budding yeast at meiotic entry and prophase, allowing synchronous release into meiosis I, and subsequently meiosis II (*Berchowitz et al., 2013*; *Carlile and Amon, 2008*; *Chia and van Werven, 2016*). These systems have been used extensively, but the short time window between the divisions limits the ability to resolve pure meiosis I and II populations. Therefore, direct comparison of meiosis I and II has remained difficult to achieve in any system.

To overcome these challenges, we sought to establish a method that relies on inducible inhibition of the Anaphase Promoting Complex (APC-Cdc20) in either meiosis I or meiosis II. The APC-Cdc20 is a ubiquitin ligase that targets key substrates for degradation, including securin and cyclin B, which is required for progression into anaphase (*Peters, 2006*). The requirement for the APC-Cdc20 at both meiosis I and II is well-established, and placement of the gene for the APC activator, *CDC20*, under control of a meiotically-repressed promoter has been widely used to arrest budding yeast cells in metaphase I (*Lee and Amon, 2003*). This system has also been adapted to allow re-expression of *CDC20* and synchronous meiosis II entry (*Argüello-Miranda et al., 2017*) as well as re-expression of the *cdc20-3* temperature-sensitive allele to enable collection of metaphase II cells (*Mengoli et al., 2021*). However, the reliance on transcriptional control and temperature shift results in a slower response than desirable. Therefore, no tools currently exist to quickly and reliably enrich budding yeast cell populations in meiosis II.

The APC-Cdc20 is a natural target of the spindle assembly checkpoint (SAC), a signalling cascade that prevents anaphase when chromosomes are not properly attached to the spindle (*McAinsh and Kops, 2023*). The SAC is initiated at improperly attached kinetochores when the kinase Mps1 phosphorylates the kinetochore protein Spc105[KNL1]. This phosphorylation recruits the checkpoint proteins Bub3/Bub1 and then Mad1/Mad2 to the kinetochore. Within this kinetochore complex, Mad2 is converted from the open to closed conformation and C-Mad2 associates with Mad3[BubRI], Bub3, and Cdc20 to form the mitotic checkpoint complex (MCC), a diffusible inhibitor of the anaphase-promoting complex (APC). Sustained SAC activation, resulting in prometaphase arrest, can be induced experimentally by perturbing kinetochore-microtubule interactions, for example, by treatment with microtubule-depolymerizing drugs, and this method has been used extensively in mitotic cells. However, meiotic cells respond poorly to such drugs, likely due to a combination of impaired uptake, toxicity, and a weakened checkpoint (*Hochwagen et al., 2005*; *MacKenzie et al., 2023*). Moreover, such treatments are unsuitable for analysis of kinetochore function since they ablate microtubule interactions and convert kinetochores into a signalling platform, thereby changing their composition and properties.

An alternative approach to arrest cells in either metaphase I or II would induce SAC-dependent APC-Cdc20 inhibition without perturbing kinetochores. Mechanistic studies in fission yeast and human cells have achieved inducible activation of the SAC through conditional dimerisation of Mps1 kinase and Spc105[KNL1] (*Amin et al., 2018*; *Aravamudhan et al., 2015*; *Chen et al., 2019*; *Leontiou et al., 2019*). Here, we adapted this approach to arrest cells in either metaphase I or metaphase II of meiosis. We demonstrate that induced dimerisation of Mps1 and Spc105[KNL1] engineered to lack their kinetochore-targeting domains efficiently arrests cells in either metaphase I or metaphase II,

in addition to mitotic metaphase. This synthetic SAC (SynSAC)-dependent arrest relies on the same downstream signalling cascade required for APC-Cdc20 inhibition during mitotic division, culminating in stabilisation of the anaphase inhibitor Pds1securin. We find that the SynSAC arrest is more potent in metaphase II than metaphase I, and provide evidence that protein phosphatase 1 (PP1) limits the duration of the SAC-mediated metaphase I delay. To demonstrate the utility of our SynSAC approach, we purified kinetochores from metaphase I, metaphase II, and mitotic metaphase SynSAC cells and analysed their composition and phosphorylation using mass spectrometry. This provided evidence that the relative enrichment of purified core kinetochore sub-complexes changes in meiotic metaphase I and II compared to mitotic metaphase, and that overall kinetochore phosphorylation is reduced in metaphase II. Therefore, by developing SynSAC as a meiotic interrogation tool, we have generated a comprehensive dataset of kinetochore composition and phosphorylation, a key resource to uncover the critical specialisations that direct distinct segregation events in meiosis I and II.

## Results
### A synthetic SAC prolongs metaphase I or metaphase II
We sought to activate the SAC to arrest cells in metaphase without disrupting kinetochores. Thus, we constructed a yeast strain which has an additional, ectopic copy of each of Spc105 and Mps1 lacking their kinetochore-binding domains and which can be inducibly dimerised. To temporally separate populations of cells undergoing meiosis I vs meiosis II, this strain also carried inducible *NDT80*, which allows synchronous release from prophase arrest upon addition of β-estradiol (*Carlile and Amon, 2008*). To induce dimerisation, we used the PYL and ABI tags, which dimerise in the presence of the plant hormone abscisic acid (ABA) (*Liang et al., 2011*; *Miyazono et al., 2009*). This system is particularly advantageous compared with the widely used rapamycin-inducible dimerisation system because it cannot affect endogenous mTOR signalling, nor does it require background mutations to avoid toxicity. Spc105, lacking its C-terminal kinetochore localisation domain (*Ghodgaonkar-Steger et al., 2020*; *Maskell et al., 2010*; *Petrovic et al., 2014*; *Roy et al., 2022*), and tagged with PYL, was produced from its endogenous promoter (Spc105$^{(1-455)}$-PYL). Similarly, the C-terminal kinase domain of Mps1, also under the control of its native promoter and lacking the N-terminal disordered domain which mediates its kinetochore localisation, was fused to ABI (Mps1$^{(440-765)}$-ABI) (*Figure 1A*; *Araki et al., 2010*). As expected, spore viability of this 'SynSAC' strain after prophase release was comparable to that of a strain lacking the ABI/PYL tagged truncations, indicating that meiosis was not impaired by these constructs (*Figure 1B*).

We scored spindle morphology after anti-tubulin immunofluorescence to determine cell cycle stage (*Figure 1C*). Prophase, metaphase I, anaphase I, metaphase II, anaphase II, and post-meiotic spindles appeared successively over the timecourse in both the absence and presence of ABA (*Figure 1D*). While SynSAC dimerisation did not alter characteristic spindle morphologies, it changed their distribution over time. Following prophase release in the absence of ABA, metaphase I and metaphase II cell populations peak at approximately 75 and 120 min, respectively, as determined by spindle morphology (*Figure 1E*). ABA addition at prophase release extended metaphase I by 15 min and metaphase II by 30 min (*Figure 1E*). This suggested that conditional dimerisation of Spc105$^{(1-455)}$-PYL and Mps1$^{(440-765)}$-ABI is sufficient to activate the SAC and delay cells in either metaphase I or metaphase II. To delay cells specifically at metaphase II, we added ABA at 100 min after prophase release, corresponding to the anaphase I peak according to spindle morphology, and this resulted in a pronounced (60 min) metaphase II delay (*Figure 1E*).

To confirm that the delay in cell cycle progression was dependent on the canonical spindle checkpoint cascade, we tested whether the SynSAC-induced metaphase delays required downstream SAC proteins. Deletion of *MAD2* or *MAD3* abolished the SynSAC delays in metaphase I and II, since cell cycle profiles were comparable when either ethanol or ABA were added at prophase release (*Figure 2A and B*). Similarly, SynSAC dimerisation after meiosis I did not delay cells in metaphase II in the absence of the SAC proteins (*Figure 2C*). Therefore, both meiotic metaphase delays induced by dimerisation of Mps1$^{(440-765)}$-ABI and Spc105$^{(1-455)}$-PYL are dependent on known downstream SAC components. We conclude that SynSAC can induce a delay in both metaphase I and metaphase II, in a manner requiring downstream checkpoint components.

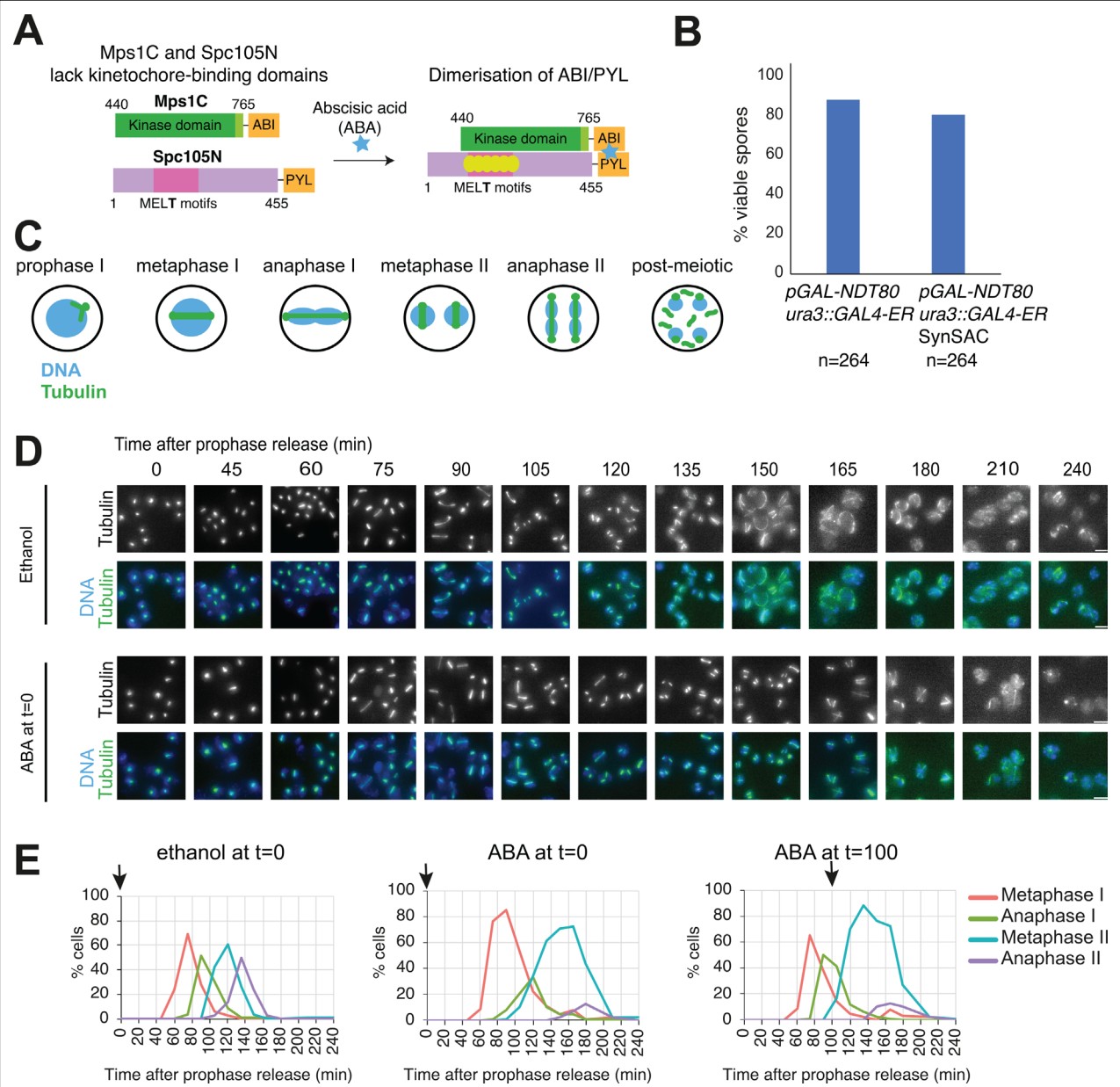

**Figure 1.** The meiotic synthetic SAC (SynSAC) activation system. (**A**) Schematic of synthetic spindle assembly checkpoint (SAC) dimerisation system. (**B**) Spore viability of wild-type (AM11189) and SynSAC dimer (AM30783) yeast strains (**C**) Schematic showing spindle morphology at different stages in meiosis. (**D and E**) SynSAC strain AM30783 was released from prophase by β-estradiol addition under control (ethanol, left), meiosis I dimerising (250 μM ABA at release), or meiosis II dimerising (250 μM ABA at 100 min) conditions. (**D**) Representative images of fields of SynSAC cells at the indicated times after prophase release. Upper panels (ethanol) show control cells where SynSAC is not activated, lower panels (abscisic acid, ABA) show cells where SynSAC is activated from prophase release. Scale bar = 5 μm. (**E**) Scoring of spindle morphology after anti-tubulin immunofluorescence. 100 cells were scored at each timepoint. Arrow indicates time of ethanol or ABA addition.

## Metaphase I, metaphase II, and mitotic metaphase show differential sensitivity to SynSAC

We noticed that SynSAC induced a more pronounced delay in metaphase II than in metaphase I (*Figures 1E and 2B and C*). Furthermore, both the metaphase I and metaphase II arrests were transient and cells subsequently underwent both meiotic divisions to produce spores, even in the presence of ABA. In contrast, SynSAC induced a robust mitotic metaphase arrest and checkpoint-dependent growth inhibition (*Figure 2—figure supplement 1A and B*). This is in accordance with findings in

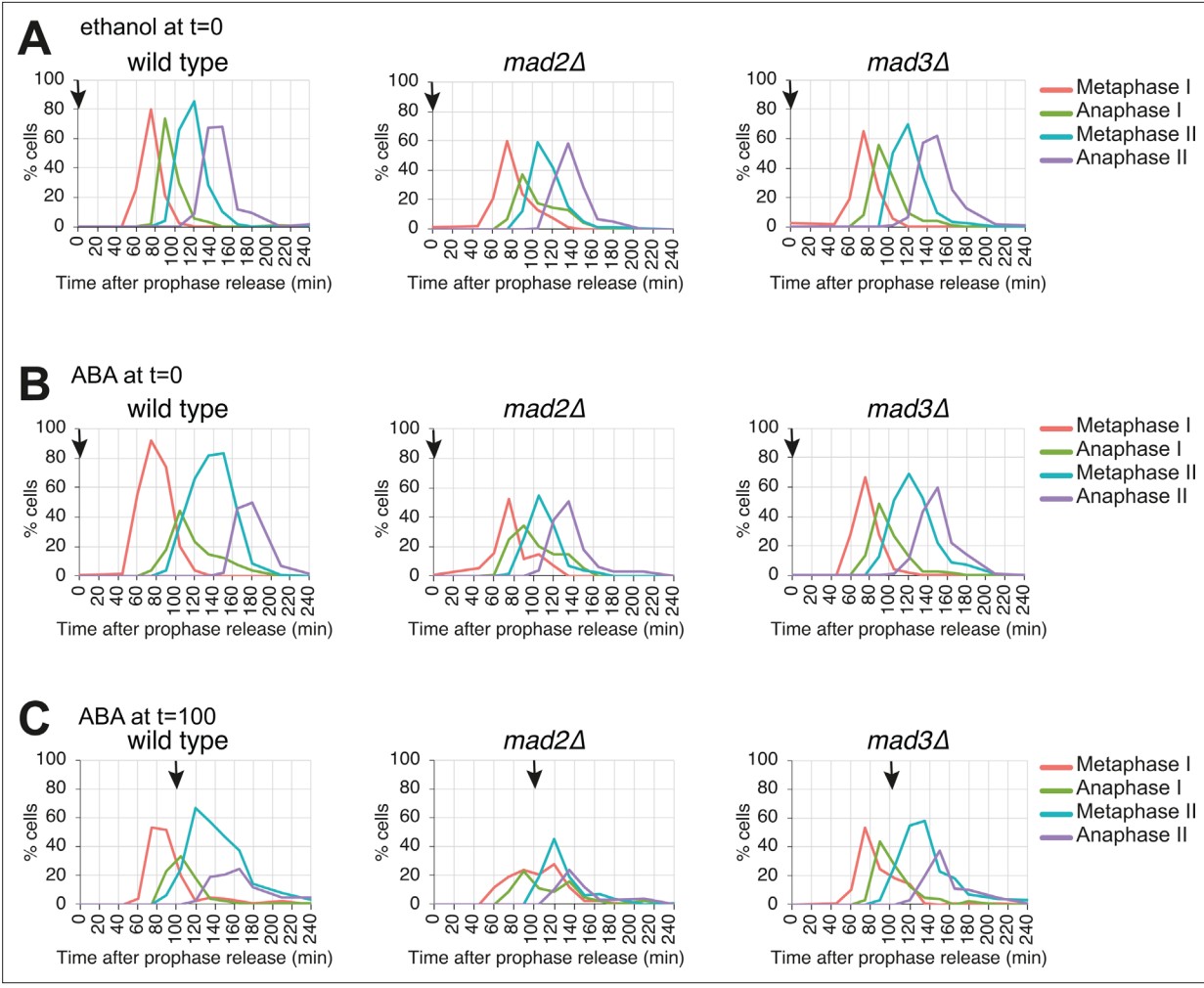

**Figure 2.** Synthetic SAC (SynSAC) activation depends on checkpoint proteins. (**A–C**) Meiosis spindle immunofluorescence timecourse in wild-type SynSAC (AM30783), SynSAC *mad2Δ* (AM33559), and SynSAC *mad3Δ* (AM30784) strains. All strains were released from prophase by β-estradiol addition. (**A**) Control cells where ethanol was added at release (**B**) SynSAC dimerisation was induced by addition of 250 μM abscisic acid at prophase release (**C**) SynSAC dimerisation was induced by addition of 250 μM of abscisic acid 100 min after release. 100 cells were scored at each timepoint.

The online version of this article includes the following figure supplement(s) for figure 2:

**Figure supplement 1.** Synthetic SAC (SynSAC) induces a robust abscisic acid (ABA)-induced mitotic arrest.

yeast and vertebrates which have suggested that the SAC is less robust in meiosis I compared to either mitotic metaphase or metaphase II (*Gui and Homer, 2012*; *Kolano et al., 2012*; *Lane et al., 2012*; *MacKenzie et al., 2023*; *Nagaoka et al., 2011*; *Sebestova et al., 2012*). However, since these previous observations rely on methods which perturb kinetochore-microtubule attachments, it has remained unclear whether this difference stems from reduced SAC signalling or a dampened response. Our finding that SynSAC, which uncouples signal generation from the response, induces a graded effect, demonstrates that meiosis I and to a lesser extent, meiosis II have a reduced response to checkpoint signalling.

To explore the differences between the meiosis I and II SAC responses further, we analysed the effects of SynSAC on downstream signalling events. The SAC inhibits the APC-Cdc20 from targeting the anaphase inhibitor securin for degradation. Therefore, we monitored Pds1[securin] degradation as a readout for APC-Cdc20 activity in the SynSAC strain. Without ABA-induced dimerisation, Pds1[securin] levels decreased along with the appearance of anaphase I spindles at 90 min after prophase release, re-accumulated at 105 min, then decreased again at 135 min when anaphase II spindles appeared (*Figure 3A*). In wild-type cells, Pds1 levels are higher in meiosis I than in meiosis II, likely because the interval between the divisions is too short to allow full Pds1 reaccumulation (*Marston et al., 2003*;

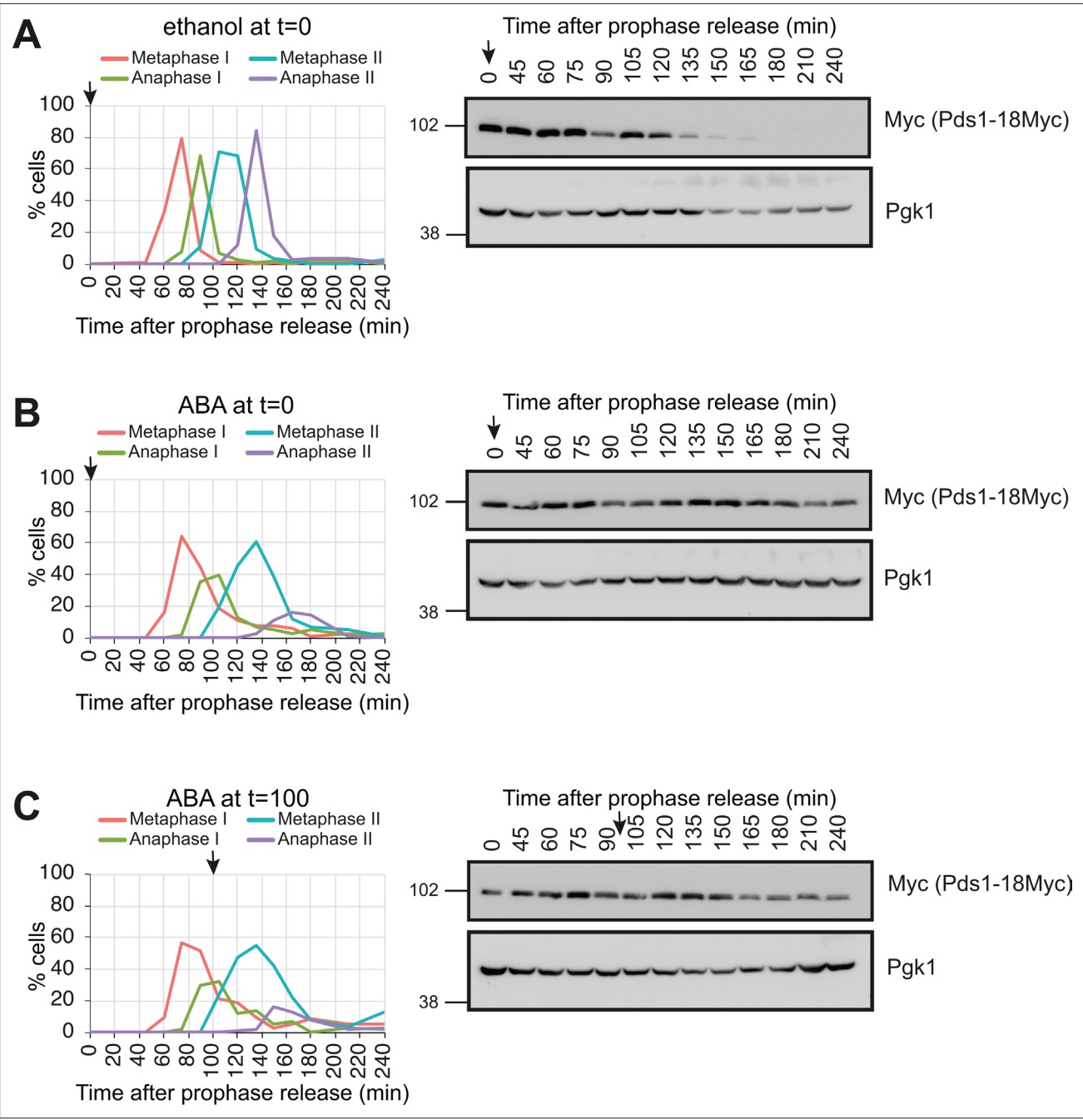

**Figure 3.** Meiotic synthetic SAC (SynSAC) delays degradation of Pds1securin. (**A–C**) Meiosis timecourse with spindle immunofluorescence (left) and western blotting to visualize Pds1-18Myc and Pgk1 loading control (right). (**A**) Control timecourse with ethanol added at prophase release. (**B**) Meiosis I SynSAC delay timecourse with 250 μM abscisic acid (ABA) added at prophase release. (**C**) Meiosis II SynSAC delay meiosis timecourse with 250 μM ABA added at 100 min after prophase release (vertical dotted line). Strain used in A-C was AM34398. For the spindle counts, 100 cells were scored at each timepoint. Arrows indicate the time of ethanol or ABA addition.

The online version of this article includes the following source data and figure supplement(s) for figure 3:

**Source data 1.** Original files for western blot data shown in *Figure 3A–C*.

**Source data 2.** Original files for western blot data shown in *Figure 3A–C* with bands and treatments annotated.

**Figure supplement 1.** Protein levels of the synthetic SAC (SynSAC) dimer constructs do not significantly change throughout meiosis.

**Figure supplement 1—source data 1.** Original files for western blot and ponceau staining data shown in *Figure 3B and D*.

**Figure supplement 1—source data 2.** Original files for western blot and ponceau staining data shown in *Figure 3B and D* with bands labelled.

**Figure supplement 2.** Synthetic SAC (SynSAC) does not affect spore viability after meiosis.

*Matos et al., 2008*; *Salah and Nasmyth, 2000*). This pattern was also observed in SynSAC strains in the absence of ABA (*Figure 3A*). Note that Pds1 levels do not fully decline in this population-based analysis, as the short duration of meiotic stages results in a mixed-stage population. For example, at the anaphase I peak (90 min), around 30% of cells remain in prior stages in which Pds1 levels are expected to be high. However, ABA addition at the time of prophase release resulted in Pds1[securin] stabilisation throughout the timecourse, consistent with delays in both metaphase I and metaphase II (*Figure 3B*). Anaphase I spindles nevertheless appeared with delayed kinetics, peaking at ~40% at 105 min. Concurrently, ~40% of cells remained in metaphase I or II and were, therefore, Pds1-positive, accounting for the persistent Pds1 signal on the western blot. In contrast, anaphase II spindles are observed at low frequency (maximum 10%) from 165 min onwards because metaphase II spindles give way to post-meiotic spindles, without undergoing anaphase II extension (*Figure 1D*). Therefore, SynSAC activation prior to metaphase I stabilises Pds1[securin] with the greatest effect in meiosis II. When ABA was added after metaphase I at 100 min after release, Pds1[securin] levels similarly remained high until the end of the timecourse and after spindle disassembly (*Figure 3C*). Together, these findings suggest that activation of the SynSAC in meiosis delays metaphase I and II by preventing Pds1[securin] degradation. Interestingly, although SynSAC induces a longer delay in metaphase II compared to metaphase I, in both cases, accummulation of metaphase spindles is transient yet accompanied by Pds1[securin] persistence.

One possibility for the extended delay in metaphase I compared to metaphase II could be differential protein expression in the two divisions. To test this, we assessed the levels of the SynSAC dimerising proteins by Western blotting. Both full-length endogenous and the C-terminal SynSAC fragment of Mps1 were tagged with 3V5 for direct comparison, which revealed increased expression of Mps1[(440-765)]-ABI-3V5 compared to Mps1-3V5 (*Figure 3—figure supplement 1A* and B), consistent with the presence of a degron in the N-terminal domain of Mps1 (*Palframan et al., 2006*). However, levels of the SynSAC Mps1[(440-765)]-ABI-3V5 fragment were relatively consistent throughout meiosis I and II (*Figure 3—figure supplement 1A and B*). Similarly, FLAG-tagging of both full-length endogenous Spc105 and Spc105[(1-455)]-PYL indicated that protein levels were equivalent across the two divisions (*Figure 3—figure supplement 1C and D*). Thus, temporal differences in the abundance of the SynSAC dimerising constructs cannot account for the longer delay in metaphase II compared to metaphase I.

Finally, the transience of the SynSAC delay at both metaphase I and metaphase II raised the question of whether escape into anaphase I and II was associated with accurate chromosome segregation. To test this, SynSAC strains were allowed to complete sporulation in the presence of ABA, and spore viability was measured. If kinetochores or chromosome segregation were significantly disrupted by the delayed meiotic cell cycle, spores would inherit the incorrect number of chromosomes and be inviable. However, spore viability after SynSAC dimerisation, either before metaphase I or specifically in metaphase II, was not significantly different from wild-type (*Figure 3—figure supplement 2*). Therefore, SynSAC does not disrupt chromosome segregation and provides a powerful tool to study meiotic metaphase.

## PP1 binding to Spc105 limits SAC-induced metaphase delay in meiosis

Why does SynSAC induce a more potent arrest in metaphase II compared to metaphase I? One possibility is that checkpoint silencing mechanisms are especially active in meiosis I and contribute to a rapid reversal of the checkpoint activity. A key event in checkpoint silencing is the docking of Protein Phosphatase 1 (PP1; Glc7 in yeast) onto two consensus motifs in the N-terminal region of Spc105, leading to the dephosphorylation of Spc105 and the disassembly of the SAC signalling platform (*Rosenberg et al., 2011*; *Roy et al., 2019*). Analysis of mitotic cells indicated that the RVxF motif is the primary PP1 docking motif and that the SILK motif (GILK in budding yeast) plays an auxiliary role (*Rosenberg et al., 2011*; *Roy et al., 2019*). To test whether PP1 binding to Spc105 regulates the duration of the SynSAC delay in meiosis, we created strains in which the dimerising copy of Spc105[(1-455)]-PYL in the SynSAC strain background had mutations in the RVxF and SILK motifs that would prevent PP1 binding. We mutated either the SILK motif, creating *spc105[(1-455)]–4A-PYL*, or the RVSF motif, creating *spc105[(1-455)]-RASA-PYL*, and additionally generated *spc105[(1-455)]–4A-RASA* which has both PP1-binding motifs mutated. We also made the *spc105[(1-455)]-RVAF* mutant in which serine 77 within the RVSF motif is mutated to alanine, to block potential Ipl1[Aurora B] phosphorylation of this site.

In human cells, Aurora B-dependent phosphorylation of KNL1[Spc105] within the RVSF motif appears to prevent PP1 binding (*Liu et al., 2010*). However, the phospho-null RVAF mutant does not impact growth of budding yeast, suggesting it may not have as much importance in this organism (*Rosenberg et al., 2011*; *Roy et al., 2019*).

In the absence of ABA, mutation of the PP1 binding sites in the SynSAC construct did not affect meiotic progression or spore viability (*Figure 4—figure supplement 1A and B*), nor did they compromise mitotic growth (*Figure 4—figure supplement 1C*). However, activating the SynSAC by adding ABA at the time of prophase release resulted in a metaphase I delay of varying duration depending on the status of the PP1-binding motifs (*Figure 4A*). Mutation of the SILK motif (*spc105(1-455)–4 A*) did not detectably alter the 15–30 min metaphase I delay observed for the wild-type SynSAC (*Figure 4A*), consistent with the notion that this motif is less important. However, preventing PP1 binding to the RVxF motif (*spc105(1-455)-RASA*) extended the duration of the SynSAC-induced metaphase I delay considerably and mutation of both PP1 binding motifs (*spc105(1-455)–4A-RASA*) led to an even more pronounced delay, with metaphase I cells persisting until the end of the timecourse (*Figure 4A*). Conversely, mutation of the potential Ipl1[Aurora B]-dependent kinase site (*spc105(1-455)-RVAF*), which is predicted to increase PP1 binding and thereby promote checkpoint silencing, modestly decreased the duration of the metaphase I delay (*Figure 4A*). Taken together, these findings indicate that PP1 binding to Spc105 dampens the SynSAC response in meiosis I and that both PP1 binding motifs play a role, with the RVxF motif being the primary binding site.

To test whether the duration of the metaphase II delay is also affected by the ability of PP1 to bind Spc105, we repeated the above experiment except ABA was added only at the time of anaphase I. As for metaphase I, preventing PP1 binding to Spc105 increased the duration of SynSAC-induced metaphase II, with the different mutations showing a gradient of effects (*Figure 4B*). The strongest metaphase II delay was observed with the *spc105(1-455)–4A-RASA* construct, with the *spc105(1-455)-RASA* mutant alone also extending metaphase II compared to the wild-type, as judged by the absence of anaphase II cells. Mutation of the SILK motif (*spc105(1-455)–4A*) caused a modest additional 15 min delay. However, we observed no detectable effect of mutating the potential Ipl1[Aurora B] site on the SynSAC metaphase II delay (*Figure 4B*). Overall, these results indicate that PP1 binding to the RVxF motif, and to a lesser extent the SILK motif, on Spc105 affects the duration of the metaphase to anaphase transition in both meiosis I and II. Interestingly, the effect on metaphase I appears more striking, potentially suggesting increased PP1-dependent SAC silencing activity in meiosis I. These findings also indicate that *spc105(1-455)–4A-RASA* is the preferred SynSAC variant, particularly where metaphase I arrest is the goal. However, due to concurrent investigations, below we employed the simple SynSAC system and demonstrated its suitability for functional studies.

## Kinetochore composition changes at meiotic prophase, metaphase I, and metaphase II

The SynSAC system we developed allows robust isolation of populations of both metaphase I and metaphase II cells from the same strain, providing the first opportunity for biochemical comparisons between these stages. As proof of principle, we analysed kinetochores, which must be modified to achieve distinct functions during meiosis. Indeed, previous work has established remodelling of kinetochore composition between meiotic prophase and meiosis I (*Borek et al., 2021*; *Meyer et al., 2015*). However, the lack of a suitable arrest system precluded analysis of meiosis II. Since kinetochores undergo a configuration switch from mono-oriented in meiosis I to bi-oriented in meiosis II to direct the successive segregation of homologs and sister chromatids, direct comparison of meiosis I and meiosis II kinetochores is an important goal. We exploited our newly developed SynSAC system to determine kinetochore composition at metaphase I and metaphase II, together with mitotic metaphase and meiotic prophase for comparison. To do so, we purified kinetochores via immunoprecipitation of the central kinetochore protein Dsn1 and used mass spectrometry to analyse their composition (*Akiyoshi et al., 2010*; *Borek et al., 2021*; *Sarangapani et al., 2014*). Metaphase arrests were confirmed by spindle immunofluorescence (*Figure 5—figure supplement 1A*) and silver staining indicated broadly similar purifications across replicates and cell cycle stages (*Figure 5—figure supplement 1B*).

To determine kinetochore composition, we used data-independent acquisition mass spectrometry (DIA-MS). Due to the high sensitivity of this method, thousands of proteins were quantified in each of three replicate immunoprecipitations from each stage (*Figure 5—figure supplement 2A*) and

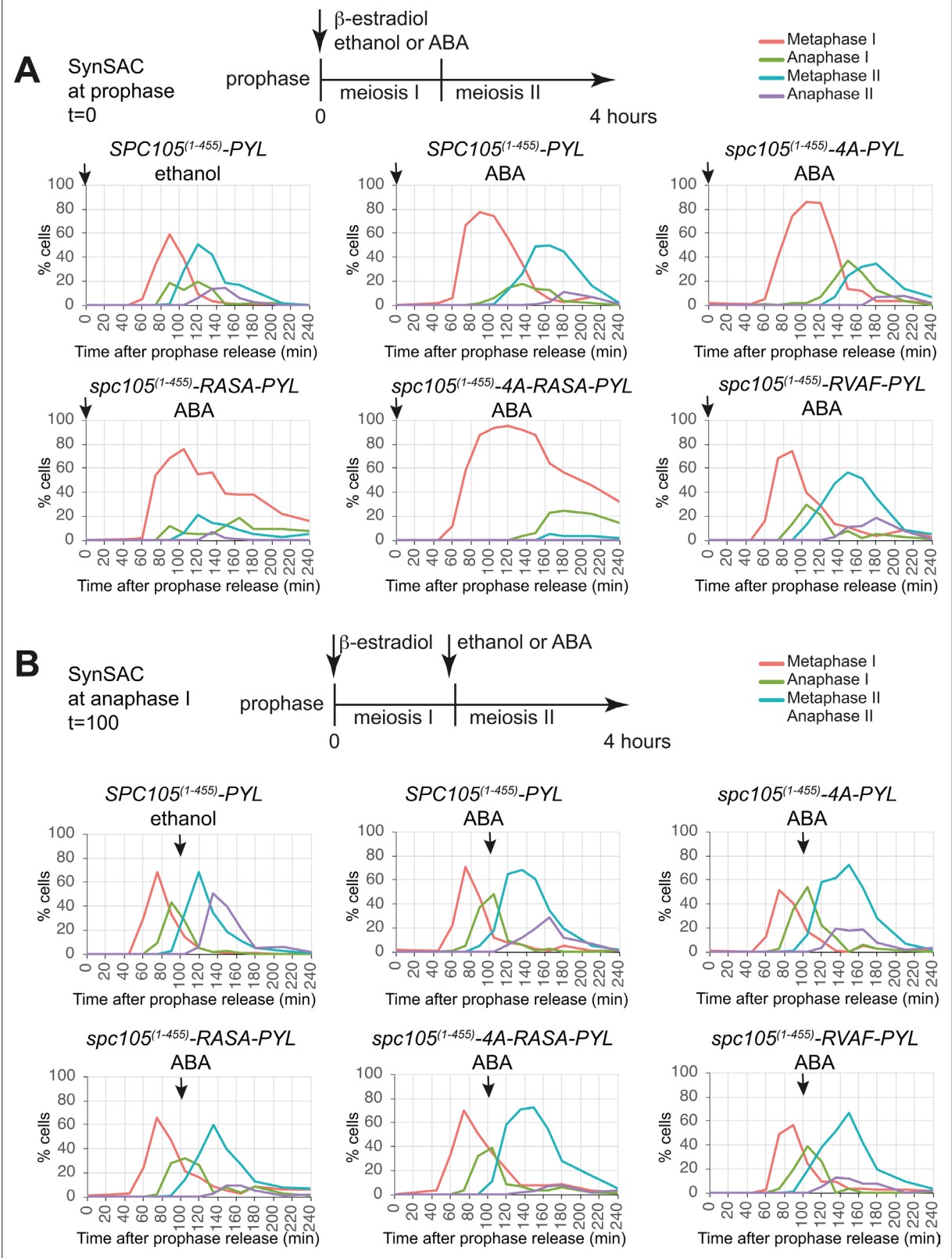

**Figure 4.** PP1 binding restrains synthetic SAC (SynSAC) delay duration in meiotic metaphase. (**A and B**) Meiosis I (**A**) and meiosis II (**B**) SynSAC spindle immunofluorescence timecourses in wild-type vs PP1 binding site mutant SynSAC strains. Top: Schematic indicating drug addition timing. Middle row: Control wild-type (AM30783), SynSAC wild-type (AM30783), SynSAC *spc105-4A* (AM34201). Bottom row: SynSAC *spc105-RASA* (AM34203), SynSAC

*Figure 4 continued on next page*

*Figure 4 continued*

*spc105-4A-RASA* (AM34202), SynSAC *spc105-RVAF* (AM34487). Arrows indicate the time of ethanol/abscisic acid (ABA) addition. 100 cells were scored at each timepoint.

The online version of this article includes the following figure supplement(s) for figure 4:

**Figure supplement 1.** Mutation of the PP1 binding sites in the Spc105 synthetic SAC (SynSAC) construct does not affect timing of meiosis, spore viability, or mitotic growth.

2415 quantified proteins were common to all three replicates of the Dsn1-His-FLAG IPs from all 4 cell cycle stages (*Figure 5—figure supplement 2B*). Consistent trends in kinetochore protein abundance were observed across replicates (*Figure 5—figure supplement 2C*). As expected, Dsn1 and other components of the core kinetochore structure, especially other members of the KMN (KNL1$^{Spc105}$-Mis12$^{MIND}$-Ndc80) network were most enriched compared to the untagged control immunoprecipitations (*Figure 5A–E*). Spindle pole body and microtubule-associated proteins were enriched in mitotic metaphase, meiotic metaphase I and metaphase II, but depleted from prophase kinetochore purifications, along with outer kinetochore (Mis12$^{MIND}$-Ndc80) complexes (*Figure 5B–E*; *Borek et al., 2021*; *Meyer et al., 2015*).

To better quantify differences in kinetochore protein abundances between the different cell cycle stages, we scaled all protein abundances to the level of Dsn1 quantified in each individual sample (*Figure 6—figure supplement 1*). As expected, the abundance of most kinetochore proteins was slightly lower than the bait protein Dsn1 in Dsn1-6His-3FLAG ('tag') samples, but not in the Dsn1 negative control immunoprecipitations ('no tag') (*Figure 6—figure supplement 1D*). Overall, there was an increased amount of quantified protein in the prophase and metaphase I kinetochore ('tag') samples compared to the metaphase II and mitotic metaphase samples (*Figure 6—figure supplement 1E*). Although this difference was not significant when the levels of known kinetochore proteins were compared (*Figure 6—figure supplement 1F*), it is possible that it reflects increased stability and/or dynamics of protein-protein interactions with the kinetochore in prophase and metaphase I.

Next, we performed pair-wise comparisons to identify which proteins, relative to Dsn1, were the most different in abundance between cell cycle stages (*Figure 6*). In addition, we used GO term analysis to identify shared functions among the top 50 non-kinetochore proteins associated with the Dsn1 immunoprecipitate at each stage (*Figure 6—figure supplement 2A*). At prophase, proteins involved in chromosome pairing, recombination, and respiration were associated with kinetochores (*Figure 6—figure supplement 2A and B*). At metaphase I, chromosome organisation, spindle pole body, and spindle organisation proteins were enriched. The enrichment of many chromatin-associated factors e.g., histones and chromatin remodeling complexes with metaphase I, but not metaphase II, kinetochores (*Figure 6—figure supplement 2A and B*) is consistent with closer association of kinetochores with chromatin in the mono-oriented vs bi-oriented state. At metaphase II, sporulation and spore wall proteins became enriched (*Figure 6—figure supplement 2A and B*). Mitotic kinetochores were associated with nucleolar, ribosome, and nuclear envelope proteins (*Figure 6—figure supplement 2A and B*). Together, these associations underscore the dynamic nature of kinetochore protein interactions. Furthermore, it suggests that the kinetochore may serve as a platform to direct cell cycle stage-specific signalling, such as promoting recombination at meiotic prophase, specialised kinetochore-spindle associations at metaphase I and instigating meiotic-exit processes, such as spore wall formation in metaphase II.

Beyond identifying novel protein interactions, we aimed to determine whether there were significant changes in the kinetochore composition itself at the different stages, relative to Dsn1. Across the four stages, we were able to quantify ~70 known kinetochore proteins (*Figure 7A*). As expected, proteins within the KMN network, of which Dsn1 is a part, were most abundant, while proteins at the inner or outer edges of the complex, or only transiently associated, were less abundant. Comparing protein abundances grouped by sub-complex or function at each stage indicated that inner kinetochore CCAN proteins were reduced at meiotic metaphase I and II compared to prophase or mitotic metaphase, although the trend was not significant, while outer kinetochore Dam1 complex proteins showed the reverse trend (*Figure 7A–B*). It is possible that meiotic kinetochores are enriched in outer kinetochore complexes to facilitate the challenge of capturing homologs in meiosis I. Alternatively, inner kinetochore protein interactions may be weakened in the two meiotic metaphases, while outer kinetochore interactions are better preserved during the purification. As expected, there was

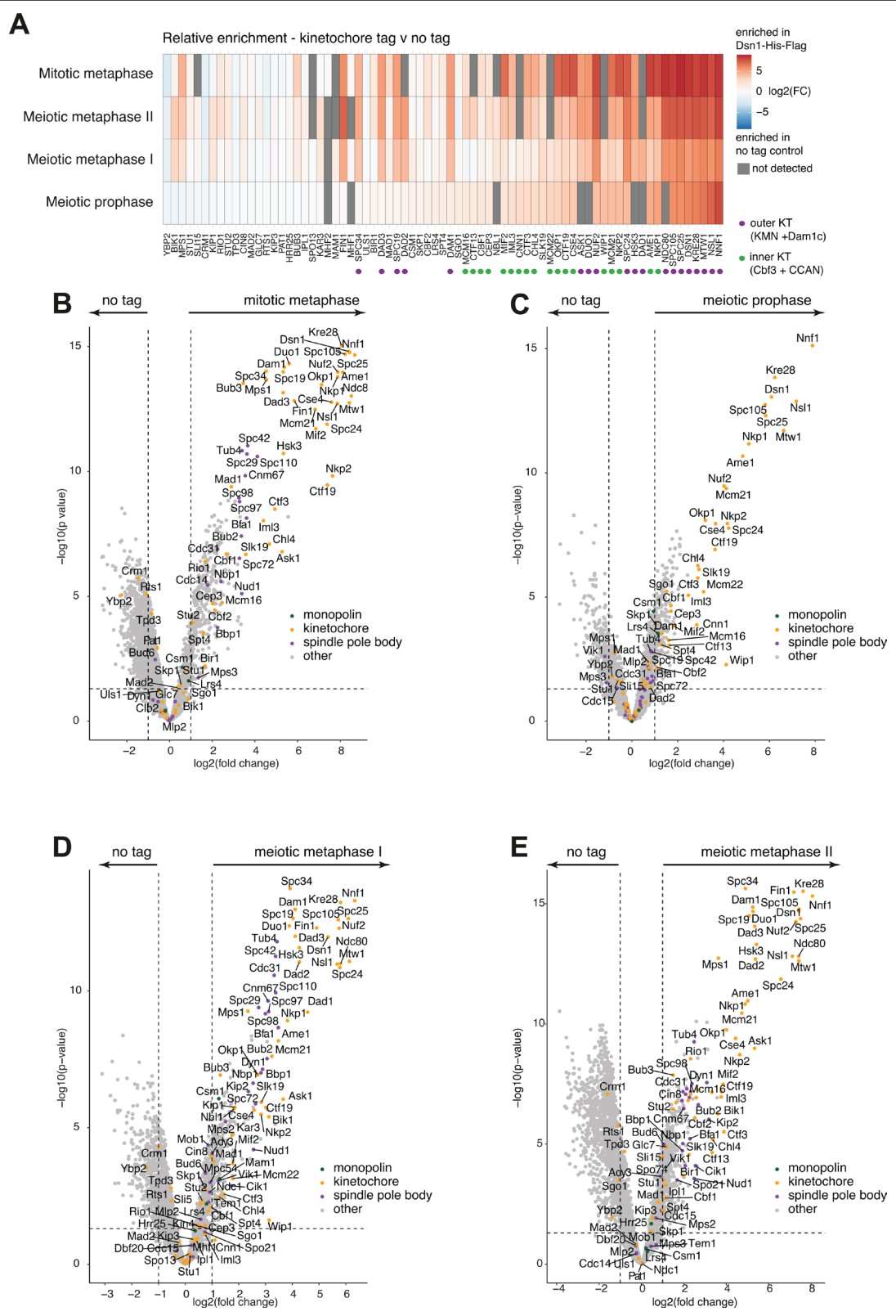

**Figure 5.** Kinetochore composition in mitotic metaphase, meiotic prophase, meiotic metaphase I, and meiotic metaphase II. (**A**) Heatmap showing the fold change enrichment level of each kinetochore protein at each stage, for the tag vs no tag comparison. (**B–D**) Volcano plots comparing unscaled protein levels of Dsn1-6His-3FLAG (tag) to no tag after anti-FLAG immunoprecipitation from matched extracts of mitotic metaphase (**B**), meiotic prophase (**C**), metaphase I (**D**), and metaphase II (**E**). Strains used are no tag (AM30990) or *DSN1-6His-3FLAG* (AM33675).

*Figure 5 continued on next page*

*Figure 5 continued*

The online version of this article includes the following source data and figure supplement(s) for figure 5:

**Figure supplement 1.** Immunoprecipitation of kinetochores from meiotic prophase, metaphase I, metaphase II, and mitotic metaphase.

**Figure supplement 1—source data 1.** Original files for silver-stained gels shown in B.

**Figure supplement 1—source data 2.** Original files for silver-stained gels shown in B with position of BSA and conditions labelled.

**Figure supplement 2.** Kinetochore composition in meiosis.

---

a significant depletion of the Ndc80 and Dam1 complex proteins from kinetochores in prophase, in agreement with previous observations (*Chen et al., 2020*; *Hayashi et al., 2006*; *Meyer et al., 2015*; *Miller et al., 2012*; *Figure 7A–B*).

Within the diverse kinetochore 'accessory' group, the strongest changes were an enrichment of Glc7$^{PP1}$ phosphatase regulator Fin1 at meiotic metaphase I and II, as well as an enrichment in Glc7$^{PP1}$ phosphatase at prophase. Additionally, proteins with known functions in mono-orientation, such as monopolin complex proteins Csm1 and Mam1 had higher abundance or were exclusively detected in metaphase I but not metaphase II kinetochores. SAC and CPC protein abundance on kinetochores was largely comparable in all stages, other than a depletion in Mps1 levels at prophase, which likely reflects the fact that this is the only condition in which SynSAC was not induced. Finally, microtubule-associated proteins (MAPs) were relatively depleted from mitotic kinetochores, and the Bik1$^{CLIP-170}$ microtubule plus-end tracking protein was particularly enriched on metaphase I and metaphase II kinetochores, consistent with the general increase in abundance of the microtubule-binding components of the outer kinetochore in these purifications (*Figure 7A–B*). Together, our results reveal that although the levels of most core kinetochore proteins are relatively stable at different cell cycle stages, there are some differences. Notably, there is a loss of outer kinetochore proteins at prophase, a depletion in some inner kinetochore protein interactions at meiotic metaphase vs mitotic, and an increase in outer kinetochore protein levels at metaphase I, and also to some extent at metaphase II.

## Reduced phosphorylation of kinetochores at metaphase II

Protein composition changes are only partially responsible for the distinct behaviour of kinetochores in meiosis I and meiosis II. Indeed, multiple kinases are known to contribute (*Clyne et al., 2003*; *Lee and Amon, 2003*; *Lo et al., 2008*; *Matos et al., 2008*; *Petronczki et al., 2006*), indicating that phosphorylation is a key mechanism driving these changes. However, a global picture of phosphorylation changes at the kinetochore is lacking. This is challenging because phospho-peptides are relatively low in abundance and recovery of sufficient quantities of protein from highly synchronised cells is especially important to detect changes between stages. Our SynSAC approach makes this possible because large quantities of highly synchronised cells can be collected within a single experiment.

To this end, 95% of the eluates from the kinetochore purifications above were subjected to phospho-peptide enrichment and DIA-MS. Thus, alongside quantification of the kinetochore-associated proteome as described above, we obtained matched phospho-proteomic datasets for each sample. Overall, we identified 4480 phosphorylation sites ('phospho-sites') on 1614 proteins with 99% confidence. The abundance of each phospho-site was normalised to the total abundance of the corresponding protein quantified in the non-phospho-enriched fraction of the sample. This more accurately identifies dynamic phosphorylation, rather than protein abundance. Additionally, we filtered the dataset to analyse only those phospho-sites which were identified in all replicates of at least one sample type. With these metrics, we quantified ~1500 phosphorylation sites on meiotic prophase and meiotic metaphase I kinetochores and ~700–1100 phospho-sites on meiotic metaphase II and mitotic metaphase kinetochores (*Figure 8—figure supplement 1A*). As expected, many phosphorylation sites were stage-specific, with 177 sites identified in all replicates of all stages (*Figure 8—figure supplement 1B*).

We performed GO term enrichment analysis to look for shared functions among the phospho-proteins most abundant in the kinetochore purifications. This was done with the top 50 most abundant phospho-proteins which are not known kinetochore proteins, in order to identify alternative processes. At prophase, phospho-proteins involved in chromosome organisation, double-strand break repair, and transcription were associated with kinetochores (*Figure 8—figure supplement 1C*). Similarly, metaphase I phospho-proteins were involved in chromatin remodeling, chromosome organisation,

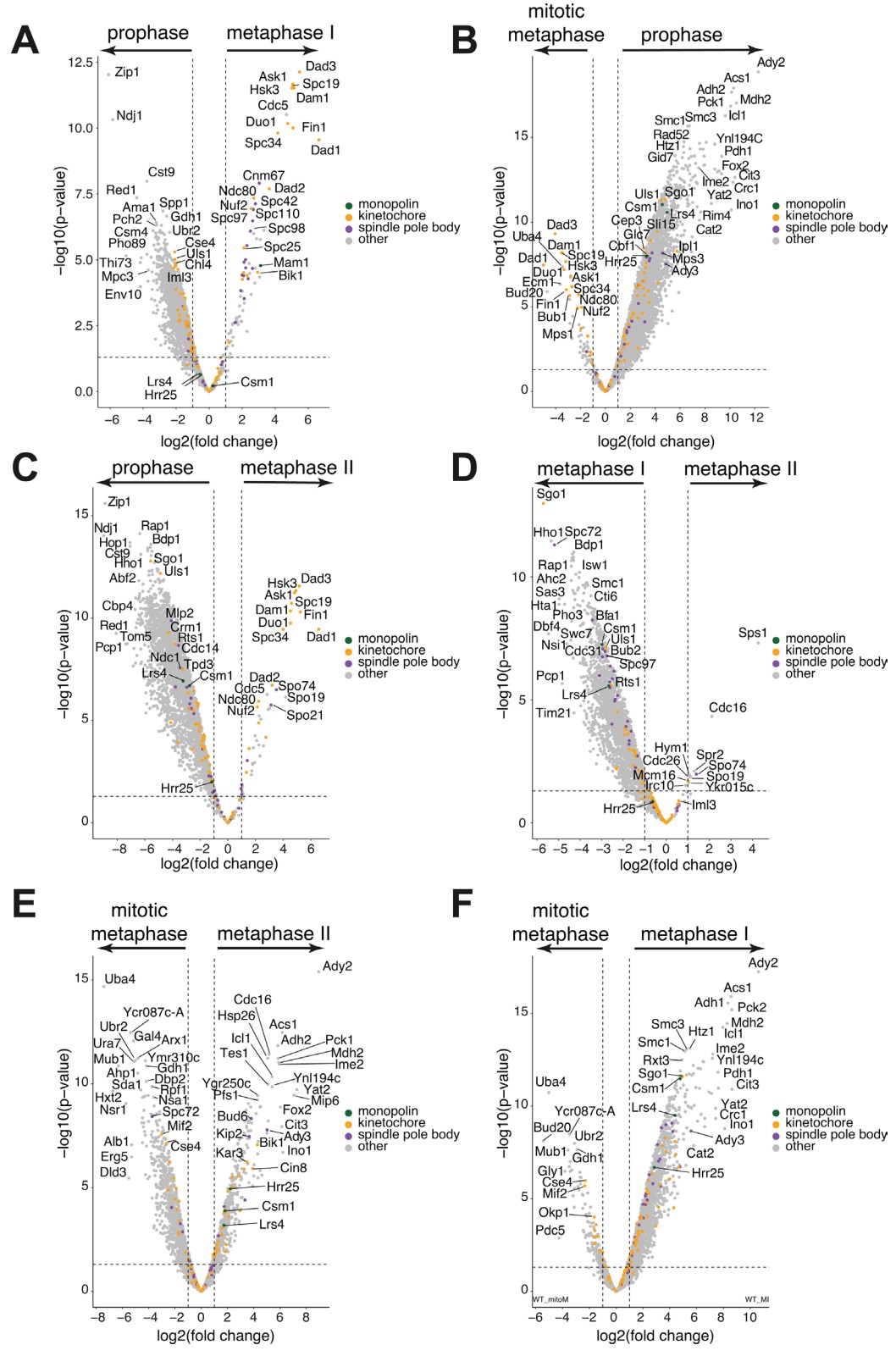

**Figure 6.** Volcano plots of differential protein enrichment for each pair-wise comparison between different cell cycle stages. Comparisons of protein abundance detected by mass spectrometry in kinetochore purifications at the different stages. (**A**) Volcano plot comparing Dsn1-scaled protein levels in meiotic prophase (left) vs metaphase I (right). (**B**) Volcano plot comparing Dsn1-scaled protein levels in mitotic metaphase (left) vs meiotic prophase

*Figure 6 continued on next page*

*Figure 6 continued*

(right). (**C**) Volcano plot comparing Dsn1-scaled protein levels in meiotic prophase (left) vs metaphase II (right). (**D**) Volcano plot comparing Dsn1-scaled protein levels in metaphase I (left) vs metaphase II (right). (**E**) Volcano plot comparing Dsn1-scaled protein levels in mitotic metaphase (left) vs metaphase II (right). (**F**) Volcano plot comparing Dsn1-scaled protein levels in mitotic metaphase (left) vs metaphase I (right).

The online version of this article includes the following figure supplement(s) for figure 6:

**Figure supplement 1.** Total protein and kinetochore protein abundances before and after scaling to Dsn1.

**Figure supplement 2.** Identity and function of proteins enriched on kinetochores at distinct stages.

and gene expression. At metaphase II, phospho-proteins involved in cell cycle regulation, sexual reproduction, and spindle pole body organisation were enriched (*Figure 8—figure supplement 1C*). Finally, mitotic metaphase phospho-proteins were involved in the microtubule cytoskeleton, nucleus organisation, and regulation of transcription. Overall, the temporal trends seen in the GO term analysis are apparent in ranked lists of the most abundant non-kinetochore phospho-sites for each stage comparison (*Figure 8—figure supplement 1D*).

To determine whether there were general trends in the total level of phosphorylation, we analysed the distribution of kinetochore phospho-site abundances at the four different stages (*Figure 8*). The number of phospho-sites on known kinetochore proteins was around ~100 at each stage, however, the median level of phosphorylation was higher in meiotic prophase and metaphase I than it was in metaphase II and mitotic metaphase (*Figure 8A*). This suggests that kinetochores are subject to essential phosphorylation in meiosis I and that either phosphatase activity increases or kinase activity towards kinetochores decreases at metaphase II. Interestingly, the total abundance of phosphorylation on kinetochores was very similar between metaphase II and mitotic metaphase, suggesting that the increased level of kinetochore phosphorylation is characteristic of meiosis I.

We next asked whether these phosphorylation events were focused on specific kinetochore subcomplexes or functional groups. The CCAN and KMN subcomplexes contained the most phosphorylation sites, while relatively few sites were mapped to the Cbf3 complex and MAPs (*Figure 8A*). The levels of phosphorylation on the CCAN and KMN subcomplexes were highest in prophase and metaphase I and lower in metaphase II and mitotic metaphase, mirroring the trend seen for all kinetochore sites. Interestingly, there appeared to be increased phosphorylation on outer kinetochore Dam1 complex proteins in metaphase I vs II, and overall higher phosphorylation in meiosis vs mitosis. Since Dam1 complex proteins are known targets of error correction kinases Ipl1$^{Aurora B}$ and Mps1, this may suggest increased error correction in meiosis vs mitosis. To identify which kinetochore proteins were most phosphorylated, we calculated the sum of the abundance of all phosphorylation sites on individual kinetochore proteins at each stage and then ranked the proteins based on the total sum from all stages (*Figure 5B*). This revealed that Spc105$^{KNL1}$, Bir1$^{Survivin}$, Lrs4, Sli15$^{INCENP}$, Dsn1$^{DSN1}$, Okp1$^{CENP-Q}$, and Ame1$^{CENP-U}$ were the most heavily phosphorylated kinetochore proteins. Additionally, we determined the maximum range in abundance between cell cycle stages for each phospho-site within each protein. This revealed that Mif2$^{CENP-C}$, Dsn1$^{DSN1}$, Cin8$^{Kif11}$, Okp1$^{CENP-Q}$, and Cse4$^{CENP-A}$ have highly dynamic phosphorylation sites (*Figure 8C*).

Finally, we highlight several individual phospho-sites with interesting trends (*Figure 8D*). Slk19$^{CENP-F}$ regulates both spindle stability and mitotic/meiotic exit (*Buonomo et al., 2003*; *Marston et al., 2003*; *Stegmeier et al., 2002*; *Sullivan et al., 2001*). We found that phosphorylation of Slk19-S-23, in part of the protein involved in meiotic exit (*Havens et al., 2010*), is reduced in metaphase II. In mitosis, phosphorylation of the N-terminus of Ndc80$^{NDC80}$ by Ipl1$^{Aurora B}$ promotes detachment of kinetochores from microtubules, allowing for correction of erroneous attachments (*Akiyoshi et al., 2009*; *Pinsky et al., 2006*; *Sarangapani et al., 2013*); in meiosis, Ipl1$^{Aurora B}$ phosphorylation of an overlapping set of sites, including T54, destabilizes Ndc80$^{NDC80}$ protein levels in prophase, contributing to the loss of outer kinetochore proteins at this time (*Chen et al., 2020*). Consistent with this, we observed significantly more phosphorylation of Ndc80-T-54 on purified kinetochores in meiotic prophase and metaphase I (*Figure 8D*). Dsn1-S-69 is found very close to where the monopolin complex that directs mono-orientation in meiosis I is thought to bind (residues 72–110) (*Plowman et al., 2019*; *Sarkar et al., 2013*). We observed the greatest phosphorylation of Dsn1-S-69 in prophase and metaphase I, the time when kinetochores prepare for mono-orientation, suggesting it may be a functionally important

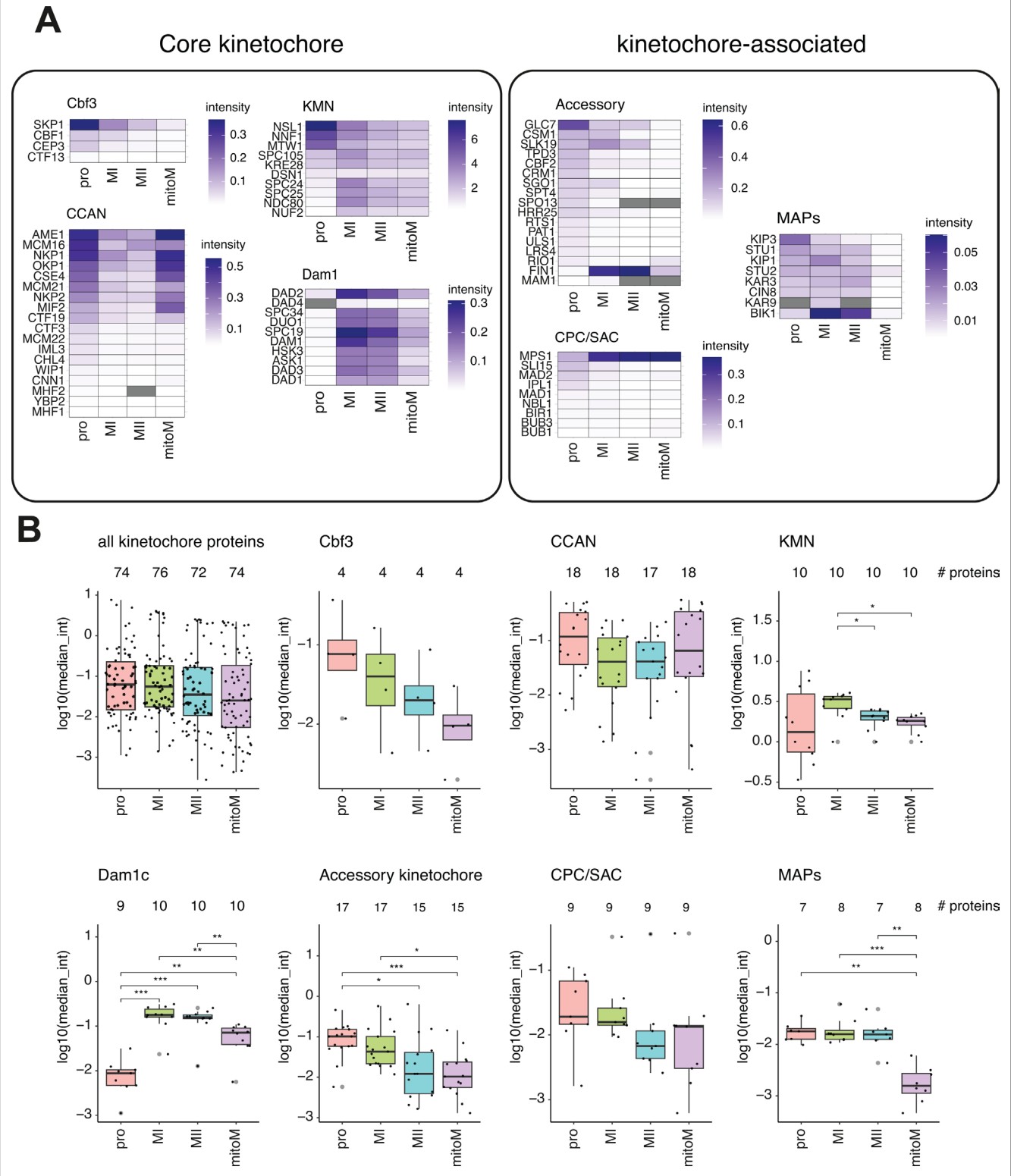

**Figure 7.** Kinetochore protein dynamics in meiotic prophase, metaphase I, metaphase II, and mitotic metaphase. (**A and B**) Kinetochores purified by Dsn1-6His-3FLAG immunoprecipitation were analysed from cells arrested at the indicated stages using the SynSAC system by mass spectrometry. Strain used was AM33675. Protein levels are Dsn1-scaled. (**A**) Heatmap of individual protein levels of core kinetochore proteins (left). Heatmap of individual protein levels of kinetochore-associated proteins at each stage (right). (**B**) Boxplots of groups of kinetochore proteins at each stage. Dots indicate individual proteins and the numbers above each plot indicate the number of proteins included in each group at that stage. P-values from the Wilcoxon two-sided test are shown.

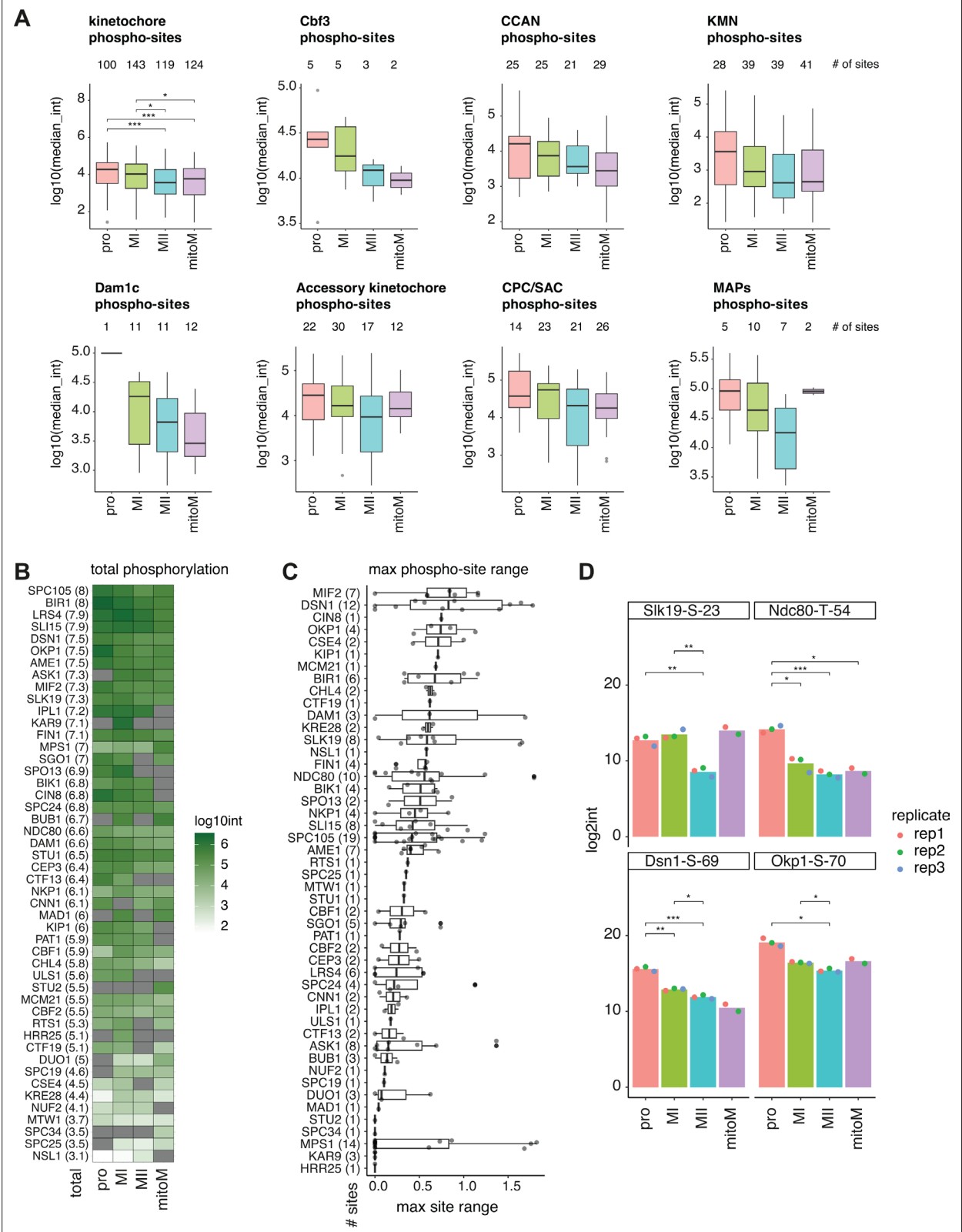

**Figure 8.** Reduced kinetochore protein phosphorylation in metaphase II. (**A–D**) Phosphorylation analysis of kinetochores purified as in *Figure 5* and subjected to phospho-enrichment prior to mass spectrometry. (**A**) Boxplots of groups of kinetochore protein phosphorylation sites at each stage. Numbers above each plot indicate the number of phospho-sites included in the group at each stage. P-values from two-sided Wilcoxon tests are shown for all kinetochore phospho-sites (upper left). No other comparisons were significantly different by the Wilcoxon test in any other group/

*Figure 8 continued on next page*

*Figure 8 continued*

stage. (**B**) Heatmap of total sum of phospho-site abundance for each kinetochore protein at each stage. Phospho-proteins are ranked, with proteins with the highest sum of phospho-site abundances, for all four stages together, at the top. Numbers within parentheses next to protein name indicate the sum of phospho-site abundances for all four stages. (**C**) Boxplots of maximum phospho-site range across the four stages for each kinetochore protein. Phospho-proteins are ranked so that proteins with the highest median phospho-site dynamic range across stages are at the top. Numbers in parentheses indicate the number of phospho-sites considered to calculate maximum phospho-site range. (**D**) Bar plots of individual phospho-site abundances for the indicated sites at each of the four stages. Dots indicate the abundance in individual replicates. P-values from two-sided t-tests are indicated.

The online version of this article includes the following figure supplement(s) for figure 8:

**Figure supplement 1.** Identification of phospho-sites co-isolated with purified kinetochores in meiotic prophase, metaphase I, metaphase II, and mitotic metaphase.

modification (***Figure 8D***). Finally, we found that S70 phosphorylation on inner kinetochore protein Okp1[CENP-Q], a likely Cdk substrate (***Holt et al., 2009***), is reduced at metaphase II.

To uncover trends in types of kinase activity directed towards kinetochore proteins, we conducted motif analysis on phospho-sites at different cell cycle stages. Of the nearly 100 kinetochore phosphorylation sites identified, the most prominent trend was for a proline residue at the +1 position from the phospho-acceptor site, which is a known motif recognised by cyclin-dependent kinase (Cdk) as well as MAP kinases (***Figure 9A***). Next, we analysed how many of the kinetochore phospho-sites at each stage matched a panel of known kinase consensus motifs. The trends were weak, suggesting common types of kinase activity at the different stages, however, there was a slight preference for the strict Cdk or Ipl1[Aurora B] kinase consensus motif among prophase phospho-sites compared with the other stages (***Figure 9B***). Additionally, fewer sites matching the Ipl1[Aurora B] motif were found in meiotic metaphase I and II (***Figure 9B***), which may suggest that Ipl1[Aurora B] activity is reduced at these times. Indeed, we would anticipate that the prolonged SynSAC arrest allows for robust chromosomal alignment and minimal requirement for Ipl1[Aurora B]-dependent error correction. We previously found that, globally, Cdc5[Polo] kinase promotes phosphorylation of a more stringent motif of [DEN]×[ST]*F specifically in meiotic metaphase I (***Koch et al., 2024***). We found that this trend also applies to kinetochores since the number of phosphorylation sites on kinetochore proteins that match this motif was also highest at metaphase I (***Figure 9B***). Finally, there was no stage-specific enrichment of the motif recognised by the DDK kinase Cdc7, known to function in DNA damage repair and homologous recombination in prophase (***Figure 9B***), suggesting that its activity is widespread through mitosis and meiosis, or that other kinases can recognise these sites.

Given that the minimal Cdk and Cdc5[Polo] kinase motifs were common among kinetochore phospho-sites, we also analysed whether these sites were preferentially located within specific sub-complexes. For Cdc5[Polo] motif sites, this revealed a slight enrichment within the outer kinetochore complexes KMN and Dam1c compared with the inner kinetochore CCAN sub-complex, and this trend was most prominent at metaphase II and mitosis (***Figure 9C***). Conversely, at all stages, there was an enrichment for Cdk motif sites within the inner kinetochore CCAN complex vs the outer kinetochore KMN and Dam1 sub-complexes. Finally, we reveal the identities of individual Cdc5[Polo] and Cdk motif phospho-sites across the core kinetochore (***Figure 9D*** and E). There was a range of different trends, with some sites being phosphorylated relatively equally across stages, while others appeared enriched at specific stages. Together, this dataset provides a strong basis for identifying phosphorylation sites which may have key stage-specific roles, such as directing mono-orientation in meiosis I and biorientation in meiosis II.

## Discussion

Mitosis, meiosis I, and meiosis II involve distinct segregation events, each with specific modifications to chromosome structure and orientation that must function in the context of a unique cell cycle programme. Understanding these changes biochemically relies on the ability to harvest cells at these distinct cell cycle stages for direct comparisons. Until now, meiosis II has been particularly elusive since robust methods to arrest budding yeast cells in this stage have been lacking. However, meiosis II is of key interest: although it is often referred to as 'resembling mitosis,' there are crucial differences since it occurs directly after another M phase, meiosis I, and is followed by gamete formation. Here,

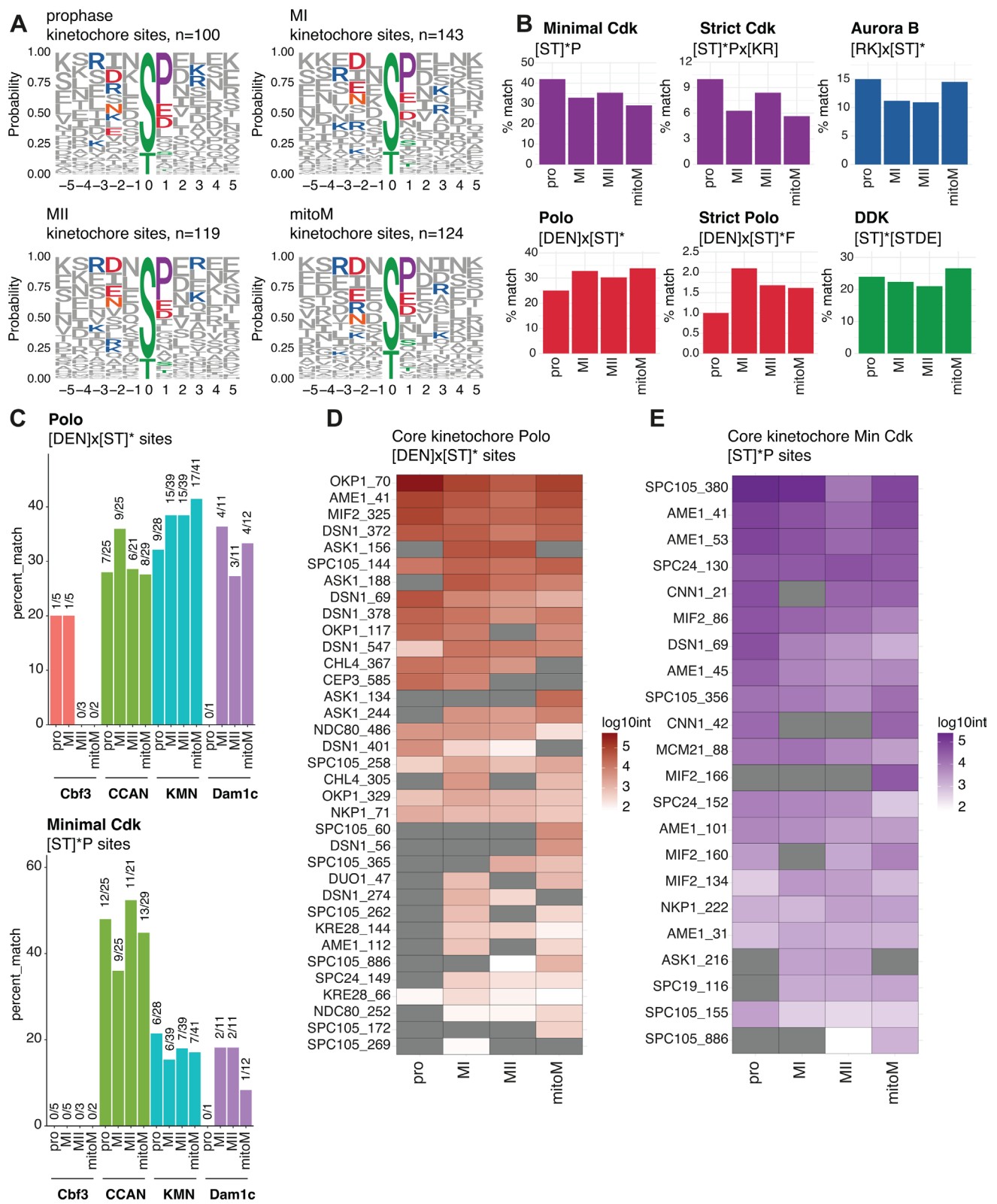

**Figure 9.** Kinetochore phospho-site motifs do not vary significantly by stage. (**A–D**) Features of kinetochore phosphorylation after phospho-analysis of kinetochore purifications in *Figure 5*. (**A**) Motif logos of amino acids surrounding kinetochore protein phospho-sites at each stage. (**B**) Bar plots of the percent of phospho-sites which match indicated kinase consensus motifs at each stage. (**C**) Bar plots of the percent of phospho-sites which match the Polo kinase consensus motif (top) or minimal Cdk consensus motif (bottom), sorted by kinetochore sub-complex and cell cycle stage. (**D**) Heatmap

*Figure 9 continued on next page*

*Figure 9 continued*

showing the abundance of individual kinetochore phospho-sites matching the minimal Polo kinase consensus. (**E**) Heatmap showing the abundance of individual kinetochore phospho-sites matching the minimal Cdk kinase consensus.

we developed the SynSAC system to overcome this limitation. SynSAC uses the same yeast strain to arrest cells in metaphase of mitosis, meiosis I, or meiosis II, allowing direct comparisons. This system could be applied to interrogate changes in any cellular process or protein complex at these distinct cell cycle stages. Here, as proof of principle, we determined the changes in protein composition and phosphorylation abundance at kinetochores. Our findings develop a new tool for meiotic investigations, provide key insights into meiotic regulation by SAC activation and additionally generate an important resource that provides key insight into how kinetochores are adapted at different cell cycle stages.

## A simple method to arrest yeast cells at metaphase II

We found that inducible dimerisation of fragments of Spc105 and Mps1 upon prophase release or during anaphase I allows for robust synchronisation at metaphase I or II of meiosis, respectively (*Figure 1*). Since the meiotic SynSAC method relies on two drug-inducible events and does not require media washout steps, it greatly simplifies and improves recovery of metaphase cells. One advantage of SynSAC is that it does not require kinetochore perturbation, as the checkpoint signal is generated ectopically. Since endogenous SAC proteins are also present, SynSAC strains can also mount a response to kinetochore perturbation, as normal. In addition, SynSAC is conditional, allowing the arrest to be induced only at the desired time. This is in contrast to previous studies which have isolated metaphase I cells through repressing transcription of *CDC20* (*Lee and Amon, 2003*), where loss of the Cdc20 protein, itself part of the mitotic checkpoint complex (*McAinsh and Kops, 2023*), may have secondary effects. Thus, by relying on relatively quick chemical reactions and achieving arrest without reducing protein levels, the SynSAC method may more accurately reflect the metaphase state. A limitation of this approach, as with any synchronisation system, is that unintended consequences of rewiring cell cycle control cannot be excluded. In the case of SynSAC, such effects could arise from sustained Bub1 signalling, from ectopic activation of the checkpoint away from kinetochores or from induction of enhanced Mps1 activity. Key findings should, therefore, be validated using orthogonal approaches.

## The duration of SAC signalling in metaphase I is regulated by PP1

Interestingly, the duration of SynSAC-mediated delay is consistently shorter in meiosis I vs II, despite equivalent protein level of the dimerising construct. Microtubule-depolymerising drugs also cause a shorter SAC delay in meiosis I compared to either meiosis II or mitosis (*MacKenzie et al., 2023*). The differential sensitivity of cells to SynSAC in meiosis I and II suggests that this is an intrinsic difference between the two divisions. Previous work has established a role for PP1 in prematurely silencing the SAC in meiosis I in the presence of improperly attached kinetochores (*MacKenzie et al., 2023*). Our work similarly implicates PP1 binding to Spc105 as being responsible for limiting the duration of the SynSAC arrest in meiosis I. We discovered that mutation of either the conserved RVxF or the GILK motif does indeed lengthen the duration of the metaphase I SynSAC arrest, with the strongest effect seen when both mutations are present. This suggests that PP1 binding to these motifs may promote Spc105 dephosphorylation and SAC silencing in metaphase I. Consistently, PP1 phosphatase Glc7 and its regulatory subunit Fin1, were enriched on meiotic kinetochores (*Figure 7A*). In contrast, we observed only minor additional delays in metaphase II when PP1 binding to Spc105 was prevented. Moreover, surprisingly, the RASA mutation, but not 4A or 4A RASA abolished the growth defect of SynSAC cells in mitosis. While this finding remains unexplained, it points to differential wiring of checkpoint activation and silencing mechanisms in mitosis, meiosis I, and meiosis II. Together, these observations suggest that PP1 activity is more prominent at metaphase I. How this relates to the unique segregation pattern in this division where homologs, rather than sister chromatids, are segregated is an important area of investigation for the future.

Previous studies in *S. cerevisiae* found no evidence that Aurora B–dependent phosphorylation of S77 within the RVSF motif inhibits PP1 binding (*Rosenberg et al., 2011*; *Roy et al., 2019*), unlike the mechanism described in human cells (*Liu et al., 2010*). In contrast, we found that the RVAF mutation

partially restores mitotic growth in the SynSAC strain and shortens the SynSAC-induced delay in metaphase I. These observations suggest that PP1 binding may nevertheless be regulated by phosphorylation within the RVSF motif to attenuate checkpoint signalling, a mechanism that may be particularly important during meiosis I. Interestingly, similar to PP1, CPC subunits, including Ipl1$^{AuroraB}$, and SAC proteins are enriched on kinetochores in metaphase I. We suggest that the physical linkage of homologs by chiasmata in meiosis I creates additional constraints on biorientation, thereby necessitating repeated error correction through opposing CPC/Aurora B and PP1 activity.

Finally, the stronger and more prolonged SynSAC arrest obtained using the PP1 binding site mutant *spc105$^{1-455}$-4A-RASA* prompts its consideration as an alternative tool for future studies, particularly where meiosis I arrest is important. At the time of performing the kinetochore immunoprecipitations, these mutations were not yet available but, as we have demonstrated, wild-type SynSAC protein fragments nevertheless yielded sufficiently enriched populations of metaphase I and II cells to allow reliable detection of stage-specific kinetochore proteins and phosphorylations. Going forward, however, we consider SynSAC-4A-RASA to be the optimal tool for inducing metaphase arrests.

## Kinetochore composition dynamics in meiosis reveal distinct functional states

We applied SynSAC to understand changes in composition of the kinetochore by quantitative proteomic analysis. Prior work used Dsn1 kinetochore immunoprecipitations and minichromosome purifications to compare kinetochore and centromere composition in mitosis, prophase I and metaphase I (*Borek et al., 2021*). However, analysis of purified metaphase II kinetochores was previously inaccessible and our application of high-sensitivity DIA mass spectrometry, together with matched phosphorylation analysis, provides a more complete picture overall. Our dataset of kinetochore composition from four stages: (1) mitotic metaphase I; (2) meiotic prophase I; (3) metaphase I; and (4) metaphase II, therefore, greatly extends our understanding of how kinetochores change during meiosis. Consistent with remodelling reported in our earlier study (*Borek et al., 2021*), we reveal a change in the stoichiometry of sub-complexes relative to each other, dependent on cell cycle stage. For example, the inner kinetochore CCAN is less abundant at meiotic metaphase I and II compared with mitosis and prophase (*Figure 7A*). The mechanisms underlying the lower abundance of CCAN proteins, which notably include some of the most essential DNA-contacting proteins, such as the Cse4$^{CENP-A}$ histone and the Ame1$^{CENP-C}$/Okp1$^{CENP-Q}$ heterodimer, during meiotic metaphase require further investigation. Moreover, it is possible that this phenomenon contributes to the essential requirement for some CCAN proteins for kinetochore assembly in meiosis, but not mitosis (*Borek et al., 2021*; *Fernius and Marston, 2009*; *Mehta et al., 2014*). Conversely, we observed higher levels of Dam1 complex proteins at meiotic metaphase I and II compared with mitosis, and this trend was significant for the complex as a whole (*Figure 7A–B*). Along the same lines, there was a small but statistically significant increase in the levels of KMN proteins in metaphase I specifically (*Figure 7A–B*). Overall, there is a trend for a loss in inner kinetochore protein interactions and a gain in outer kinetochore protein interactions in meiotic metaphase compared to mitotic metaphase. Although some variation could reflect global changes in protein abundance during meiosis, we previously found that only a few proteins undergo dynamic abundance changes during the meiotic divisions (*Koch et al., 2024*), so this is unlikely to fully explain the kinetochore composition differences observed. Instead, it is tempting to speculate that kinetochores are more loosely associated with chromatin and more strongly associated with the spindle through the outer kinetochore in meiotic vs mitotic metaphase. Potentially, kinetochore-spindle interactions are stabilised to better ensure accurate segregation during the rapid metaphase-anaphase transitions of meiosis and to cope with the challenging configuration of homologs in meiosis I, relying on chiasmata-dependent linkages.

Our data also provide insight into the unique regulatory state of meiosis I kinetochores. Consistent with our previous findings (*Borek et al., 2021*), meiotic axis (Red1) and synaptonemal complex proteins (Zip1) are enriched on prophase kinetochores and the meiosis I-specific monopolin complex was enriched on metaphase I kinetochores. Cohesins (Smc1 and Smc3) were also enriched on metaphase I kinetochores, supporting the requirement for a dedicated pool of centromeric cohesion for mono-orientation at this stage (*Barton et al., 2022*). Casein kinase 1 (CK1/Hrr25), which plays multiple roles in meiosis, including as a component of monopolin (*Argüello-Miranda et al., 2017*; *Galander et al., 2019*; *Petronczki et al., 2006*), was immunoprecipitated with kinetochores at all meiotic stages.

As noted above, PP1 phosphatase and CPC-Aurora B kinase enrichment further suggest dynamic phospho-regulation in metaphase I consistent with our detection of increased kinetochore-associated phosphorylations at this stage compared to metaphase II or mitosis (*Figure 8A*). Furthermore, the pericentromeric cohesin protector, shugoshin (Sgo1), was highly enriched on metaphase I compared to metaphase II kinetochores. This is consistent with recent work in mouse oocytes which indicated that shugoshin-dependent cohesin protection is lost prior to meiosis II (*El Jailani et al., 2025*). Nevertheless, imaging experiments show that shugoshins are retained at meiosis II kinetochores (*Galander et al., 2019*; *Katis et al., 2004*; *Marston et al., 2004*; *Mengoli et al., 2021*), albeit at a reduced level, suggesting that they may play additional roles in meiosis II. Overall, these findings indicate dynamic changes in the composition of kinetochore-associated proteins during meiosis and provide a starting point for future studies directed at their functional significance.

## Metaphase II kinetochores have reduced phosphorylation

Our phospho-proteomic analysis of purified kinetochores from meiosis and mitosis identified 4487 phosphorylation sites on 1614 proteins, indicating that ~35% of the detected proteome is phosphorylated, in agreement with previous studies (*Figure 8—figure supplement 1A*; *Ptacek et al., 2005*; *Wettstein et al., 2024*). Nearly twice as many phospho-sites were identified in meiotic prophase and metaphase I compared with metaphase II and mitotic metaphase (*Figure 8—figure supplement 1A*). Around 250–400 more proteins were detected in prophase and metaphase I, suggesting unique regulation at the level of protein abundances as well as phosphorylation (*Figure 5—figure supplement 2A*). We found less variation in the number of phospho-sites found on known kinetochore proteins, with around 100 sites identified on ~70 kinetochore proteins across all stages (*Figure 8A*). Despite this similarity, we quantified significant differences in the overall abundances of the phospho-sites, with meiotic prophase and metaphase I kinetochores having significantly higher levels of phosphorylation compared with meiotic metaphase II and mitotic metaphase (*Figure 8A*). Within the core kinetochore structure, most phospho-sites mapped to the chromatin-associated CCAN or the spindle-associated KMN complexes. Around 40% of CCAN phosphosites matched the CDK consensus, suggesting that CDK control of kinetochore assembly timing may be concentrated at the inner kinetochore. It is also interesting that the two most dynamic phosphoproteins identified in our study, Dsn1 and Mif2$^{CENP-C}$ (*Figure 8C*), are implicated in recruitment of factors required for the unique mono-oriented arrangement of sister chromatids in meiosis I. An N-terminal extension on Dsn1 directly binds monopolin (*Plowman et al., 2019*; *Sarkar et al., 2013*), while mouse and fission yeast CENP-C recruit MOKIRs Moa1 and Meikin (*Kim et al., 2015*; *Tanaka et al., 2009*). It therefore,seems highly likely that some of the stage-specific phosphorylation events are important for regulating the switch in kinetochore orientation in meiosis, and future research should aim for their functional characterisation.

Furthermore, most of the detected phosphorylation sites were unique to the stage and, therefore, their function is likely to be stage-specific (*Figure 8—figure supplement 1B*). Whether sister kinetochore biorientation at metaphase II and mitotic metaphase is physically and/or biochemically distinct remains unclear. Our characterisation of meiotic SynSAC duration and previous work with the endogenous SAC (*MacKenzie et al., 2023*) suggests that metaphase II is regulated differently, with metaphase II having a shorter arrest compared to mitotic metaphase upon SynSAC activation. However, previous biophysical characterisation indicated that the kinetochore-microtubule attachment strength of metaphase II and mitotic purified kinetochores was equivalent on average (*Sarangapani et al., 2014*). Thus, it may be that the shorter duration of metaphase II does not depend on maximum attachment strength per se and rather on some intrinsic cell cycle regulation or other properties of kinetochores in these stages. Indeed, the distance between sister centromeres in mouse metaphase II oocytes is nearly double that in human and mouse mitotic cells, so there are likely biophysical differences between biorientation at these two stages (*Kouznetsova et al., 2019*). Additionally, as many as ~20% of chromosomes in mouse metaphase II cells appear to have lateral or merotelic attachments that persist into anaphase II, when they are then corrected (*Kouznetsova et al., 2019*). Thus, compared with mitosis, error-correction mechanisms are notably different in meiosis. Error correction relies on phospho-regulation of the outer kinetochore sub-complexes KMN and Dam1c, and we observed higher levels of Dam1 complex proteins in kinetochores from meiosis vs mitosis. This, in addition to the differential phospho-sites identified on outer kinetochore proteins, provides a good

first step for future investigations into meiotic kinetochore attachment mechanisms and how they differ in meiosis I and meiosis II.

## Materials and methods

### Plasmids

Plasmids used in this study are listed in *Supplementary file 1*.

*MPS1(440–765)* in a *HIS3* single integration plasmid AMp1725 was made by restriction digest of empty *HIS3* single integration plasmid AMp1694 (received from Biggins Lab) and *MPS1(440–765)* amplified from wild-type SK1 yeast genomic DNA. *pMPS1-MPS1(440–765)* in a *HIS3* single integration plasmid AMp1740 was made by Gibson assembly of ~500 bp of *MPS1* promoter amplified from wild-type SK1 yeast genomic DNA and AMp1725. *pMPS1-MPS1(440–765)-AB*I in a *HIS3* single integration plasmid AMp1755 was made by Gibson assembly of ABI amplified from AMp1460 (received from Hardwick Lab, made in Heun Lab) and AMp1740.

*pSPC105-SPC105(1-455)-PYL-3FLAG* in a *LEU2* single integration plasmid AMp1741 was made by Gibson assembly of three parts. ~500 bp of *SPC105* promoter and *SPC105(1-455)* were amplified from wild-type SK1 genomic DNA, *PYL-3FLAG* was amplified from plasmid AMp1461 (received from Hardwick Lab, made in Heun Lab) and *LEU2* single integration plasmid AMp1697 (received from Biggins Lab).

*pSPC105-SPC105(1-455)-PYL* in a *LEU2* single integration plasmid AMp2055 was made by megaprimer mutagenesis (two-step PCR) to completely remove the 3FLAG tag in AMp1741.

PP1 binding site mutations in *SPC105-PYL* plasmids were made by megaprimer mutagenesis. *spc105(1-455)–4A* mutates the 21-GILK-24 motif to 21-AAAA-24. *spc105(1-455)-RASA* mutates the 75-RVSF-78 motif to 75-RASA-78. The *spc105(1-455)–4A-RASA* plasmid contains both sets of mutations. *spc105(1-455)-RVAF* mutates serine 77 to alanine only.

### Yeast strains

*Saccharomyces cerevisiae* strains used in this study were derivatives of SK1 and are listed in *Supplementary file 2*. *pGAL1-NDT80 pGPD1-GAL4.ER* was described previously (*Benjamin et al., 2003*). The *MPS1-3V5*, *SPC105-6His-3FLAG*, and *DSN1-6His-3FLAG* strains were made by standard PCR tagging methods. SynSAC strains with *MPS1(440–765)-ABI* and *SPC105(1-455)-PYL* integrated at the *his3* and *leu2* locus, respectively, were made by restriction digestion of the appropriate plasmids and standard yeast DNA transformation procedures. Yeast strains are available upon request without restriction.

### Meiotic prophase block-release timecourse

Cells were induced to undergo meiosis as described by *Barton et al., 2022* and *pGAL-NDT80* prophase block-release experiments were performed as outlined in *Carlile and Amon, 2008*. Briefly, strains were patched from −70 °C stocks to YPG agar (1% Bacto yeast extract, 2% Bacto peptone, 2.5% glycerol, 0.3 mM adenine, 2% agar) plates. After ~16 hr, cells were inoculated into YPDA media and grown for 24 hr at 30 °C with shaking at 250 rpm. Next, BYTA (1% Bacto yeast extract, 2% Bacto tryptone, 1% potassium acetate, 50 mM potassium phthalate) cultures were prepared to $OD_{600} = 0.3$ and grown at 30 °C with shaking at 250 rpm overnight (~16 hr). The next morning, cells were washed twice in sterile water and resuspended in sporulation medium (0.3% potassium acetate, pH = 7.0) at $OD_{600} = 2.0$. After 5.5 hr in sporulation media, 1 μM β-estradiol was added to release cells from prophase and samples were collected for immunofluorescence every 15 min until 180 min, and then every 30 min for a further hour. For experiments where protein samples were analysed by Western blotting, 10 ml of meiotic culture or 5 ml of mitotic culture was collected by centrifugation and resuspended in 5 ml ice-cold 5% trichloroacetic acid (TCA). Cells were collected again by centrifugation and transferred to Fastprep (MP) tubes. Cell pellets were snap-frozen in liquid nitrogen and stored at −70 °C. Frozen cell pellets were washed in 500 μl acetone and air-dried. Cell pellets were resuspended in TE with 2.75 μM DTT and 1 x Roche complete protease inhibitor cocktail. Protein samples were prepared by bead-beating in an MP Fastprep machine, at speed 6.5 for three rounds of 45 s with 1 min on ice in between, at 4 °C. SDS Sample buffer was added (final concentration 3% SDS) and samples were boiled for 3 min before PAGE.

## Spore viability assay

All spore viability assays were carried out following prophase block-release timecourses. Following the timecourse, cultures were left shaking at 30 °C~24 hr. 100 µl of culture was collected by spinning in a microcentrifuge for 1 min at 13 k rpm and then resuspended in 20 µl 1 mg/ml zymolyase in 1 M sorbitol and incubated for 10 min at room temperature before adding 1 mL sterile water. Tetrads were dissected with a micromanipulator on a Nikon Eclipse 50i light microscope.

## Mitotic alpha factor arrest-release timecourse

Yeast strains were patched from −80 °C stocks to YPG agar (1% Bacto yeast extract, 2% Bacto peptone, 2.5% glycerol, 0.3 mM adenine, 2% agar) plates. The next day, strains were patched to YPDA (1% Bacto yeast extract, 2% Bacto peptone, 2% glucose, 0.3 mM adenine) agar plates. The next day, cells were inoculated into YPDA medium and grown overnight ~16 hr at 30 °C with shaking at 250 rpm. The next morning, cultures were diluted to OD ~0.2 and grown for 2 hr at room temperature with shaking at 250 rpm. Cultures were diluted back to OD ~0.2 and alpha factor was added to 7.5 µg/ml. Cultures were shaken at room temperature for 3 hr before checking that >90% of cells were shmooing or unbudded. Cells were washed three times in 2 x volume water and collected by centrifuging in a tabletop centrifuge at 3 k rpm for 2 min at room temperature. Cells were resuspended in fresh YPDA and placed into clean flasks. Culture samples of 100 µl were taken at time zero before addition of either ethanol or 250 µM abscisic acid (ABA) and then at 30 min intervals after addition of ethanol or ABA. 100 µl culture samples were immediately combined with 400 µl ethanol and stored at 4 °C. For DNA staining, cells were pelleted and resuspended in 100 µl 1 µg/ml DAPI in PBS. Yeast cells were sonicated for 30 s on low in a Bioruptor Twin sonicating device (Diagenode) and glass slides were prepared and visualised on a Zeiss Axioplan Imager Z2 fluorescence microscope with a 100 x Plan ApoChromat NA 1.45 oil lens. At least 100 cells were counted for each condition.

## Serial dilution mitotic growth assay

Yeast strains were inoculated into YPDA medium and grown overnight ~16 hr at room temperature. Cultures were diluted to $OD_{600}$=0.2 and grown to mid-log at room temperature before being diluted back to $OD_{600}$=0.3. Serial dilutions were set up in 96-well plates, with fivefold dilutions across rows. Yeast were transferred to YPDA plates or YPDA plates spread with 250 µM abscisic acid (ABA). Plates were incubated at 30 °C for ~24 hr before taking images.

## Spindle immunofluorescence

Meiotic spindles were visualised by indirect immunofluorescence as described in *Barton et al., 2022*. A rat anti-tubulin primary antibody (Bio-Rad Cat# MCA78G, RRID:AB_325005) at 1:50 dilution and an anti-rat FITC-conjugated secondary antibody (Jackson ImmunoResearch Labs Cat# 712-095-153, RRID:AB_2340652) at 1:100 dilution were used. A Zeiss Axioplan Imager Z2 fluorescence microscope with a 100 x Plan ApoChromat NA 1.45 oil lens was used to visualise cells. In total, 100–200 cells were counted at each timepoint and/or for each sample.

## SDS-PAGE and Western blotting

SDS-PAGE and Western blotting were carried out largely as described in *Barton et al., 2022*, with minor modifications. Protein samples were separated on 10% bis-tris acrylamide gels submerged in running buffer (25 mM Tris, 190 mM glycine, 0.01% SDS) using the Bio-Rad Mini-Protean or Scie-plas TV200Y wide format mini gel system. SDS-PAGE gels were transferred onto nitrocellulose membrane (0.45 µM, Amersham-GE Healthcare) in transfer buffer (25 mM Tris, 1.5% glycine, 0.02% SDS, 10% methanol) in a Bio-Rad Mini Trans-Blot system or a semi-dry Amersham TE70 transfer unit.

Membranes were blocked in 5% milk in PBS with 0.01% Tween-20 (PBST) for at least 30 min before incubation in primary antibody overnight with gentle shaking at 4 °C. At room temperature, membranes were washed three times for 10 min in PBST before incubation in secondary antibody for 1 hr. Membranes were washed for 10 min, three times, before visualisation using chemiluminescence. Anti-rabbit and anti-mouse HRP-conjugated antibodies (Cytiva Cat# NA934, RRID:AB_772206; Cytiva Cat# NA931, RRID:AB_772210) were detected with ECL blotting substrate (Pierce), SuperSignal West Pico Plus substrate (Pierce), or SuperSignal West Atto substrate (Pierce) for weaker signals and images were acquired with a ChemiDoc MP Imaging System (Bio-Rad). Primary antibodies used were mouse

anti-Myc (Agilent Cat# 257261, RRID:AB_10640570) and rabbit anti-Pgk1 (laboratory stock *Fox et al., 2017*).

## Growth conditions for kinetochore immunoprecipitations

All immunoprecipitations were performed with lysate from either a *DSN1-6His-3FLAG* strain (AM33675) or no tag *DSN1* (AM30990) strain grown in different conditions.

For each replicate from meiotic cultures, two cultures of 400 mL $OD_{600}$ ~2.0 were sporulated in 4 L flasks and scored for metaphase arrest by spindle IF as appropriate. Cultures that were most similar by date of growth and spindle IF were combined during later grinding lysis. For each replicate from mitotic cultures, 800 ml $OD_{600}$ = 1.1–1.3 culture was grown in one 4 L flask and harvested and processed together.

For each 400 mL meiotic culture, strains were patched from –70 stocks to YPG plates and grown at 30° C. After ~24 hr, a 25 mL YPDA culture in a 250 mL flask was inoculated and grown at 30° C, shaking at 250 rpm. After ~24 hr, a 200 mL BYTA culture in a 2 L flask was started at $OD_{600}$ = 0.2 and grown at 30° C, shaking at 250 rpm. The next morning, after ~16 hr, a 400 ml culture in a 4 L flask was started at $OD_{600}$ ~2.0 in sporulation medium. All meiotic cultures were grown at 30° C with shaking at 250 rpm. For meiotic prophase, cultures were incubated for 5.5 hr before harvest. For metaphase I, cultures were incubated for 5.5 hr before 1 μM β-estradiol and 250 μM ABA were added together. After 90 min, cells were harvested. For metaphase II, cultures were incubated for 5.5 hr before 1 μM β-estradiol was added. After 105 min, 250 μM ABA was added and cells were harvested at 135 min. Samples for spindle immunofluorescence were collected at the time of harvest.

For each mitotic culture, strains were patched from –70 stocks to YPG plates and grown at 30° C. After ~24 hr, a 25 mL YPDA culture was inoculated in a 125 mL flask and grown at 30° C with shaking at 250 rpm. After ~16 hr, an 800 mL YPDA culture was started in a 4 L flask at $OD_{600}$ ~0.3 and grown at 30° C with shaking at 250 rpm. After ~3 hr, the $OD_{600}$ = 1.1–1.2 culture was harvested.

## Immunoprecipitation of kinetochores for mass spectrometry

Immunoprecipitations were carried out essentially as described in *Borek et al., 2021*, with minor modifications.

### Harvest

Cell cultures were spun at 4 k rpm for 5 min at 4° C in a Beckman Avanti J25 centrifuge. Each 400 ml cell culture pellet was washed in 150 mL water and spun again the same way. Then, cell pellets were resuspended in BH0.15 lysis buffer (150 mM KCl, 25 mM HEPES-KOH pH 8.0, 2 mM $MgCl_2$, 0.5 mM EGTA-KOH pH 8.0, 0.1 mM EDTA-KOH pH 8.0, 0.1% NP-40), including phosphatase inhibitors (5 mM NaF, 2 mM Na-beta-glycerophosphate, 1 mM sodium pyrophosphate, 800 μM sodium orthovanadate, 200 nM microcystin-LR) and protease inhibitors (2 mM Pefabloc, 10 μg/ml each of 'CLAAPE' (Chymostatin, Leupeptin, Aprotinin, Antipain, Peptstatin A, E-64), and 1 x Roche cOmplete inhibitor cocktail). Cell pellets were resuspended in BH0.15, including inhibitors according to the following equation: $OD_{600}$ × culture volume in L × 2=mL buffer (i.e. for each 400 mL $OD_{600}$ = 2.0 pellet, cells were resuspended in 1.6 mL buffer). Resuspended cultures were drop-frozen into liquid nitrogen and stored at –70° C before further processing.

### Lysis

Frozen cell pellets were lysed by cryo-grinding in a SPEX 6875 Freezer Mill, with 5 min pre-cool, 2 min Run time, 2 min Cool time, 8 cycles of 10 cycles per second. Cryo-grindate was stored at –70° C before further processing.

### Antibody conjugation to Dynabeads

Monoclonal mouse anti-FLAG antibody (Sigma-Aldrich Cat# F3165, RRID:AB_259529) was conjugated to Dynabeads as described in *Borek et al., 2021*. Antibody-conjugated Protein G Dynabeads (Thermo Fisher) were always prepared no more than 24 hr before use and were stored at 4° C until use.

## Immunoprecipitation and elution

Grindate powder, in 50 mL Falcons, was thawed in room temperature water before transferring to ice. Thawed grindate was clarified by centrifugation in a tabletop centrifuge at 3500 rpm for 10 min 4° C. The supernatant, the lysate, was transferred to a new Falcon tube and concentration measured by standard Bradford Assay (BioRad Cat#5000006). IPs were set up so that 80–100 mg protein was incubated with ~100 µL anti-FLAG conjugated Protein G Dynabeads (Thermo Fisher). The ratio of protein to beads was always kept such that 12.6 mg protein was incubated with 15 µL antibody-conjugated beads. IPs of replicates of the same condition were all done on the same day and normalised so that protein concentration and antibody-bead volumes were the same between replicates. Protein lysates and antibody-conjugated Dynabeads were incubated together in Falcon tubes with gentle rotation at 4° C for 3 hr. Next, supernatants were collected and beads were washed twice in 1 mL BH0.15 with phosphatase and protease inhibitors by gently inverting the tubes, then washed once in 1 mL BH0.15 alone. Protein was eluted from beads in two rounds of 10 min at 50° C in a ThermoMixer C (Eppendorf) at 500–600 rpm with 0.1% RapiGest SF in 50 mM Tris pH 8.0, added to one-half volume of Dynabeads (i.e. two rounds of 50 µl elution for 100 ul Dynabeads). The two sequential elutions were combined, and the eluate was stored at –70° C before further processing.

## Silver staining

Protein samples were separated on 4–12% NuPAGE pre-cast gels (Invitrogen) run in MES buffer (50 mM MES, 50 mM Tris Base, 0.1% SDS, 1 mM EDTA, pH 7.3) and stained using an Invitrogen Silverquest staining kit following the manufacturer's instructions.

## Preparation of samples for mass spectrometry

RapiGest eluates were prepared for mass spectrometry by trypsin digestion with a filter-aided sample preparation (FASP) method, essentially as described in *Koch et al., 2024*; *Wiśniewski et al., 2009*.

## Phospho-peptide enrichment

For each trypsin-digested peptide solution, 5% was loaded directly onto C18 stage tips, called the 'N' sample, and 95% was processed for phospho-peptide enrichment via Ti-IMAC beads (MagReSyn) as described in *Koch et al., 2024*, and called the 'PE' (for phospho-enriched) sample before loading on C18 stage tips. Samples of the same condition were all processed on the same day (e.g. all prophase samples were processed together on the same day).

## DIA mass spectrometry

The 'N' and 'PE' peptides eluted from the filter units were acidified using 20 µl of 10% Trifluoroacetic Acid (TFA) (Sigma Aldrich). Samples were spun onto StageTips as described by *Rappsilber et al., 2007*. Peptides were eluted in 40 µL of 80% acetonitrile in 0.1% TFA and concentrated down to 1 µL by vacuum centrifugation (Concentrator 5301, Eppendorf, UK). They were then prepared for LC-MS/MS analysis by diluting it to 5 µL by 0.1% TFA.

LC-MS analyses were performed on Orbitrap Exploris 480 (Thermo Fisher Scientific, UK) on a Data Independent Acquisition (DIA) mode, coupled online, to an Ultimate 3000 HPLC (Dionex, Thermo Fisher Scientific, UK). Peptides were separated on a 50 cm (2 µm particle size) EASY-Spray column (Thermo Scientific, UK), which was assembled on an EASY-Spray source (Thermo Scientific, UK) and operated constantly at 55 °C. Mobile phase A consisted of 0.1% formic acid in LC-MS grade water and mobile phase B consisted of 80% acetonitrile and 0.1% formic acid. Peptides were loaded onto the column at a flow rate of 0.3 µL min⁻¹ and eluted at a flow rate of 0.25 µL min⁻¹ according to the following gradient: 2 to 40% mobile phase B in 150 min and then to 95% in 11 min. Mobile phase B was retained at 95% for 5 min and returned back to 2% a minute after until the end of the run (160 min).

Survey scans were recorded at 120,000 resolution (scan range 350–1650 m/z) with an ion target of 5.0e6 and injection time of 20 ms. MS2 was performed in the Orbitrap at 30,000 resolution with a scan range of 200–2000 m/z, maximum injection time of 55 ms and AGC target of 3.0E6 ions. We used HCD fragmentation with stepped collision energy of 25.5, 27, and 30. We used variable isolation windows throughout the scan range ranging from 10.5 to 50.5 m/z. Narrow isolation windows (10.5–18.5 m/z) were applied from 400 to 800 m/z and then gradually increased to 50.5 m/z until the

end of the scan range. The default charge state was set to 3. Data for both survey and MS/MS scans were acquired in profile mode.

## Mass spectrometry library search conditions

The DIA-NN software platform (*Demichev et al., 2020*) version 1.9.2. was used to process the DIA raw files and search was conducted against our in-house *Saccharomyces cerevisiae* complete/reference proteome (original database from *Saccharomyces* Genome Database for the strain SK1, released in May, 2019). Precursor ion generation was based on the chosen protein database (automatically generated spectral library) with deep-learning-based spectra, retention time, and IMs prediction. The digestion mode was set to specific with trypsin allowing a maximum of two missed cleavages. Carbamidomethylation of cysteine was set as a fixed modification. Oxidation of methionine, acetylation of the N-terminus, and phosphorylation on serine, threonine, and tyrosine were set as variable modifications. The parameters for peptide length range, precursor charge range, precursor m/z range, and fragment ion m/z range, as well as other software parameters were used with their default values. The precursor FDR was set to 1%. Annotating library proteins were created with information from the FASTA database.

## Mass spectrometry data analysis in R

All data analysis was performed using the report.pg_matrix.tsv and report.phosphosites_99.tsv files from DIA-NN version 1.9.2 using R (v 4.2.3) within the RStudio environment.

For proteins, the columns corresponding to the measurements of the non-phospho-enriched ('N') samples in the report.pg_matrix.tsv file were analysed. For phospho-sites, the columns corresponding to the measurements of the phospho-enriched ('PE') samples in the report.phosphosites_99.tsv file were analysed. Only phospho-peptides with greater than or equal to 99% localisation confidence for the phospho-modification at the given site residue are included in the measurements in the report.phosphosites_99.tsv file. In DIA-NN, phospho-site abundances can include measurements from multiply-modified peptides; however, the vast majority of phospho-sites are quantified from singly phospho-peptides.

No imputation was carried out. Due to low coverage, only two out of three replicates of the mitotic Dsn1 ('no tag') 'N' samples were analysed for protein levels and only two out of three replicates from each of the mitotic Dsn1 ('no tag') and mitotic Dsn1-6His-3FLAG ('tag') 'PE' samples were analysed for phospho-site abundances. Samples from Dsn1-6His-3FLAG ('tag') IPs and Dsn1 ('no tag') IPs were normalised separately. All IP samples of the same type ('tag' or 'no tag'), including all replicates and cell cycle stages, were normalised together. For both the protein measurements of the 'N' samples and the phospho-site measurements from the 'PE' samples, a column median normalisation was carried out; all values in a given sample column were multiplied by a scaling factor consisting of the median intensity of all sample columns divided by the median sample intensity of that individual column. To normalise phospho-site abundances to the protein level, each column-normalised value from the 'PE' measurements of the phosphosites_99 file were divided by the corresponding column-normalised protein (from 'N') measurement from the pg_matrix table and multiplied by 1000. For example, the normalised value of phospho-site Ndc80-T-54 in the PE sample of the phosphosites_99 table was divided by the normalised value of the protein Ndc80 in the corresponding N sample of the pg_matrix and multiplied by 1000.

To better analyse differential protein levels between IPs, the protein measurements from the 'N' samples were further scaled by the measurement of Dsn1 within each sample. Each normalised protein measurement was divided by the normalised value of Dsn1 within that sample. All protein measurements shown in *Figures 6 and 7* are Dsn1-scaled.

Protein and phospho-site measurements were further analysed using functions from the Differential Enrichment of Proteins (DEP) R package (*Zhang et al., 2018*). Both protein and phospho-site tables were filtered to include only proteins or sites that were measured in all replicates of at least one condition.

Venn diagrams were made using https://interactivenn.net (*Heberle et al., 2015*). For the GO term enrichment analysis, ranked lists were generated. Proteins or phospho-proteins were ranked by descending fold change, and known kinetochore proteins were excluded. For each stage, comparison with the other stages generated three ranked lists. The top 50 proteins from all three lists were

combined together in one list, and duplicates were removed. Using these lists, GO term enrichment was carried out using the gprofiler2 R package (*Kolberg et al., 2020*) with organism='scerevisiae' and otherwise default settings. For the ranked lists in *Figure 6—figure supplement 2*, *Figure 8—figure supplement 1*, proteins were ranked by descending fold change, known kinetochore proteins were excluded, and the top 20 proteins were selected.

The stat_pwc() function from the ggpubr R package was used to run statistical tests comparing protein or phospho-site intensities. For boxplots, a two-sided Wilcoxon test was run and p-values adjusted with the Bonferroni method. For bar plots in *Figure 8D*, a two-sided t-test was run and p-values adjusted with the Bonferroni method. In all figures, a single asterisk (*) indicates a p-value of less than 0.05, two asterisks (**) indicate a p-value of less than 0.01, and three asterisks (***) indicate a p-value of less than 0.001.

The ggseqlogo R package was used to make phospho-site motif logos (*Wagih, 2017*).

## Acknowledgements

We are grateful to Sue Biggins, Kevin Hardwick, and Patrick Heun for plasmids, to Marina Altamirano De Castro for assistance with plasmid and strain construction, and to Flora Paldi for initial exploratory experiments for induction of metaphase II arrest. We thank Kevin Hardwick and all members of the Marston group for helpful discussions. We gratefully acknowledge the Wellcome Discovery Research Platform for Hidden Cell Biology Proteomics Core for mass spectrometry support. This work was funded through a Wellcome Investigator award to ALM [220780], a Wellcome Multi-User Equipment Grant [218305], core funding for the Wellcome Centre for Cell Biology [203149] and a Wellcome Discovery Research Platform Award [226791].

## Additional information

### Funding

| Funder | Grant reference number | Author |
|---|---|---|
| Wellcome Trust | 10.35802/220780 | Lori B Koch<br>Adèle L Marston |
| Wellcome Trust | 10.35802/218305 | Christos Spanos |
| Wellcome Trust | 10.35802/203149 | Lori B Koch<br>Christos Spanos<br>Adèle L Marston |
| Wellcome Trust | 10.35802/226791 | Christos Spanos<br>Adèle L Marston |
| Wellcome Trust | 319314 | Lori B Koch<br>Adèle L Marston |
| The Darwin Trust of Edinburgh | | Tiasha Ghosh |

The funders had no role in study design, data collection and interpretation, or the decision to submit the work for publication. For the purpose of Open Access, the authors have applied a CC BY public copyright license to any Author Accepted Manuscript version arising from this submission.

### Author contributions

Lori B Koch, Conceptualization, Data curation, Software, Formal analysis, Investigation, Visualization, Methodology, Writing – original draft, Writing – review and editing; Tiasha Ghosh, Formal analysis, Investigation, Visualization; Christos Spanos, Data curation, Formal analysis; Adèle L Marston, Conceptualization, Supervision, Funding acquisition, Visualization, Writing – review and editing

### Author ORCIDs

Lori B Koch ⓘ https://orcid.org/0009-0007-5286-9516
Tiasha Ghosh ⓘ https://orcid.org/0009-0004-2104-1853

Christos Spanos https://orcid.org/0000-0002-4376-8242
Adèle L Marston https://orcid.org/0000-0002-3596-9407

Reviewer #1 (Public review): https://doi.org/10.7554/eLife.110117.3.sa1
Reviewer #2 (Public review): https://doi.org/10.7554/eLife.110117.3.sa2
Reviewer #3 (Public review): https://doi.org/10.7554/eLife.110117.3.sa3
Author response https://doi.org/10.7554/eLife.110117.3.sa4

## Additional files

### Supplementary files

Supplementary file 1. List of plasmids used in this study.

Supplementary file 2. List of strains used in this study.

MDAR checklist

Source code 1. R script used for mass spectrometry analysis and figure generation in this study.

### Data availability

Mass spectrometry data generated in this study has been deposited at the PRIDE database with dataset identifier PXD067911. The R script for all data analysis in this study is provided as Source code 1 and published on Github (https://github.com/lori-koch/Kinetochore-SynSAC copy archived at *Koch, 2026*).

The following dataset was generated:

| Author(s) | Year | Dataset title | Dataset URL | Database and Identifier |
|---|---|---|---|---|
| Koch LB, Spanos C, Marston AL | 2025 | Specialisation of meiotic kinetochores revealed through a synthetic spindle assembly checkpoint strategy | https://www.ebi.ac.uk/pride/archive/projects/PXD067911 | PRIDE, PXD067911 |

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
