## [Editor Report · eLife Assessment]

Koch et al. describe a **valuable** novel methodology, SynSAC, to synchronise cells to analyse meiosis I or meiosis II or mitotic metaphase in budding yeast. The authors present **convincing** data to validate abscisic acid-induced dimerisation to induce a synthetic spindle assembly checkpoint (SAC) arrest that will be of particular importance to analyse meiosis II. The authors use their approach to determine the composition and phosphorylation of kinetochores from meiotic metaphase I and metaphase II that will be of interest to the broader meiosis research community.

[Editors' note: this paper was reviewed by Review Commons.]

---

## [Referee Report · Reviewer #1 (Public review)]

[Editors' note: this version has been assessed by the Reviewing Editor without further input from the original reviewers. The authors have comprehensively addressed the comments raised in the previous round of review.]

Summary:

These authors have developed a method to induce MI or MII arrest. While this was previously possible in MI, the advantage of the method presented here is it works for MII, and chemically inducible because it is based on a system that is sensitive to the addition of ABA. Depending on when the ABA is added, they achieve a MI or MII delay. The ABA promotes dimerising fragments of Mps1 and Spc105 that can't bind their chromosomal sites. The evidence that the MI arrest is weaker than the MII arrest is convincing and consistent with published data and indicating the SAC in MI is less robust than MII or mitosis. The authors use this system to find evidence that the weak MI arrest is associated with PP1 binding to Spc105. This is a nice use of the system.

The remainder of the paper uses the SynSAC system to isolate populations enriched for MI or MII stages and conduct proteomics. This shows a powerful use of the system, but more work is needed to validate these results, particularly in normal cells.

Overall, the most significant aspect of this paper is the technical achievement, which is validated by the other experiments. They have developed a system and generated some proteomics data that maybe useful to others when analyzing kinetochore composition at each division.

---

## [Referee Report · Reviewer #2 (Public review)]

Summary:

The manuscript submitted by Koch et al. describes a novel approach to collect budding yeast cells in metaphase I or metaphase II by synthetically activating the spinde checkpoint (SAC). The arrest is transient and reversible. This synchronization strategy will be extremely useful for studying meiosis I and meiosis II, and compare the two divisions. The authors characterized this so named syncSAC approach and could confirm previous observations that the SAC arrest is less efficient in meiosis I than in meiosis II. They found that downregulation of the SAC response through PP1 phosphatase is stronger in meiosis I than in meiosis II. The authors then went on to purify kinetochore-associated proteins from metaphase I and II extracts for proteome and phosphoproteome analysis. Their data will be of significant interest to the cell cycle community (they compared their datasets also to kinetochores purified from cells arrested in prophase I and -with SynSAC in mitosis).

Significance:

The technique described here will be of great interest to the cell cycle community. Furthermore, the authors provide data sets on purified kinetochores of different meiotic stages and compare them to mitosis. This paper will thus be highly cited, for the technique, and also for the application of the technique.

---

## [Referee Report · Reviewer #3 (Public review)]

Summary:

In their manuscript, Koch et al. describe a novel strategy to synchronize cells of the budding yeast *Saccharomyces cerevisiae* in metaphase I and metaphase II, thereby facilitating comparative analyses between these meiotic stages. This approach, termed SynSAC, adapts a method previously developed in fission yeast and human cells that enables the ectopic induction of a synthetic spindle assembly checkpoint (SAC) arrest by conditionally forcing the heterodimerization of two SAC components upon addition of the plant hormone abscisic acid (ABA). This is a valuable tool, which has the advantage that induces SAC-dependent inhibition of the anaphase promoting complex without perturbing kinetochores. Furthermore, since the same strategy and yeast strain can be also used to induce a metaphase arrest during mitosis, the methodology developed by Koch et al. enables comparative analyses between mitotic and meiotic cell divisions. To validate their strategy, the authors purified kinetochores from meiotic metaphase I and metaphase II, as well as from mitotic metaphase, and compared their protein composition and phosphorylation profiles. The results are presented clearly and in an organized manner.

Significance:

Koch et al. describe a novel methodology, SynSAC, to synchronize budding yeast cells in metaphase I or metaphase II during meiosis, as well and in mitotic metaphase, thereby enabling differential analyses among these cell division stages. Their approach builds on prior strategies originally developed in fission yeast and human cells models to induce a synthetic spindle assembly checkpoint (SAC) arrest by conditionally forcing the heterodimerization of two SAC proteins upon addition of abscisic acid (ABA). The results from this manuscript are of special relevance for researchers studying meiosis and using *Saccharomyces cerevisiae* as a model. Moreover, the differential analysis of the composition and phosphorylation of kinetochores from meiotic metaphase I and metaphase II adds interest for the broader meiosis research community. Finally, regarding my expertise, I am a researcher specialized in the regulation of cell division.

---

## [Author Response]

The following is the authors’ response to the original reviews

**General Statements**

We are delighted that all reviewers found our manuscript to be a technical advance by providing a much sought after method to arrest budding yeast cells in metaphase of mitosis or both meiotic metaphases. The reviewers also valued our use of this system to make new discoveries in two areas. First, we provided evidence that the spindle checkpoint is intrinsically weaker in meiosis I and showed that this is due to PP1 phosphatase. Second, we determined how the composition and phosphorylation of the kinetochore changes during meiosis, providing key insights into kinetochore function and providing a rich dataset for future studies.

The reviewers also made some extremely helpful suggestions to improve our manuscript, which we will have now implemented:

(1) Improvements to the discussion. Following the recommendation of the reviewers recommended we have focused our discussion on the novel findings of the manuscript and drawn out some key points of interest that deserve more attention.

(2) We added a new Figure 5 to help interpret the mass spectrometry data, to address Reviewer #3, point 4.

(3) We added a new additional control experiment to address the minor point 1 from reviewer #3. Our experiment to confirm that SynSAC relies on endogenous checkpoint proteins was missing the cell cycle profile of cells where SynSAC was not induced for comparison. We have performed this experiment and the new data is show as part of a new Figure 2.

(4) We included representative images of spindle morphology as requested by Reviewer #1, point 2 in Figure 1.

**Point-by-point description of the revisions**

**Reviewer #1 (Evidence, reproducibility and clarity):**
These authors have developed a method to induce MI or MII arrest. While this was previously possible in MI, the advantage of the method presented here is it works for MII, and chemically inducible because it is based on a system that is sensitive to the addition of ABA. Depending on when the ABA is added, they achieve a MI or MII delay. The ABA promotes dimerizing fragments of Mps1 and Spc105 that can't bind their chromosomal sites. The evidence that the MI arrest is weaker than the MII arrest is convincing and consistent with published data and indicating the SAC in MI is less robust than MII or mitosis. The authors use this system to find evidence that the weak MI arrest is associated with PP1 binding to Spc105. This is a nice use of the system.The remainder of the paper uses the SynSAC system to isolate populations enriched for MI or MII stages and conduct proteomics. This shows a powerful use of the system but more work is needed to validate these results, particularly in normal cells.Overall the most significant aspect of this paper is the technical achievement, which is validated by the other experiments. They have developed a system and generated some proteomics data that maybe useful to others when analyzing kinetochore composition at each division. Overall, I have only a few minor suggestions.

We appreciate the reviewer’s support of our study.

(1) In wild-type - Pds1 levels are high during M1 and A1, but low in MII. Can the authors comment on this? In line 217, what is meant by "slightly attenuated? Can the authors comment on how anaphase occurs in presence of high Pds1? There is even a low but significant level in MII.

The higher levels of Pds1 in meiosis I compared to meiosis II has been observed previously using immunofluorescence and live imaging[1–3]. Although the reasons are not completely clear, we speculate that there is insufficient time between the two divisions to re-accumulate Pds1 prior to separase re-activation. We added the following sentence at Line 218: “ In wild-type cells, Pds1 levels are higher in meiosis I than in meiosis II, likely because the interval between the divisions is too short to allow Pds1 reaccumulation [1,2,4]. This pattern was also observed in SynSAC strains in the absence of ABA (Figure 3A).

We agree “slightly attenuated” was confusing and we have re-worded this sentence to read “However, ABA addition at the time of prophase release resulted in Pds1^securin^ stabilisation throughout the time course, consistent with delays in both metaphase I and II”. (Line 225).

We do not believe that either anaphase I or II occur in the presence of high Pds1. Western blotting represents the amount of Pds1 in the population of cells at a given time point. The time between meiosis I and II is very short even when treated with ABA. For example, in Figure 2B (now Figure 3B), spindle morphology counts show that at 105 minutes, 40% of cells had anaphase I spindles (and will be Pds1 negative), while ~20% had metaphase I and ~20% metaphase II spindles (and will be Pds1 positive). In contrast, due to the better efficiency of the meiosis II arrest, anaphase II hardly occurs at all in these conditions, since anaphase II spindles (and the second nuclear division) are observed at very low frequency (maximum 10%) from 165 minutes onwards. Instead, metaphase II spindles partially or fully breakdown, without undergoing anaphase extension. Taking Pds1 levels from the western blot and the spindle data together leads to the conclusion that at the end of the time-course, these cells are biochemically in metaphase II, but unable to maintain a robust spindle. Spindle collapse is also observed in other situations where meiotic exit fails, and potentially reflects an uncoupling of the cell cycle from the programme governing gamete differentiation[3,5,6]. We re-wrote this section as follows. (Line 222).

“Note that Pds1 levels do not fully decline in this population-based analysis as the short duration of meiotic stages results in a mixed-stage population. For example, at the anaphase I peak (90 minutes) around 30% of cells remain in prior stages in which Pds1 levels are expected to be high. However, ABA addition at the time of prophase release resulted in Pds1^securin^ stabilisation throughout the time course, consistent with delays in both metaphase I and metaphase II. (Figure 3B). Anaphase I spindles nevertheless appeared with delayed kinetics, peaking at ~40% at 105 min. Concurrently, ~40% of cells remained in metaphase I or II and were therefore Pds1-positive, accounting for the persistent Pds1 signal on the western blot. In contrast, anaphase II spindles are observed at low frequency (maximum 10%) from 165 minutes onwards because metaphase II spindles give way to post-meiotic spindles, without undergoing anaphase II extension (Figure 1D).”

(2) The figures with data characterizing the system are mostly graphs showing time course of MI and MII. There is no cytology, which is a little surprising since the stage is determined by spindle morphology. It would help to see sample sizes (ie. In the Figure legends) and also representative images. It would also be nice to see images comparing the same stage in the SynSAC cells versus normal cells. Are there any differences in the morphology of the spindles or chromosomes when in the SynSAC system?

We have now included representative images as Figure 1D along with a schematic Figure 1C. This shows that there are no differences in spindle morphology or nuclei (chromosomes cannot be observed at this resolution), except of course the number of cells with a particular spindle morphology at a given time. We added the following text confirming that there is no change in spindle morphology (Line 174). “We scored spindle morphology after anti-tubulin immunofluorescence to determine cell cycle stage (Figure 1C). Prophase, metaphase I, anaphase I, metaphase II, anaphase II and post-meiotic spindles appeared successively over the timecourse in both the absence and presence of ABA (Figure 1D). While SynSAC dimerisation did not alter characteristic spindle morphologies, it changed their distribution over time.”

The number of cells scored (at least 100 cells per timepoint) is given in the figure legends.

(3) A possible criticism of this system could be that the SAC signal promoting arrest is not coming from the kinetochore. Are there any possible consequences of this? In vertebrate cells, the RZZ complex streams off the kinetochore. Yeast don't have RZZ but this is an example of something that is SAC dependent and happens at the kinetochore. Can the authors discuss possible limitations such as this? Does the inhibition of the APC effect the native kinetochores? This could be good or bad. A bad possibility is that the cell is behaving as if it is in MII, but the kinetochores have made their microtubule attachments and behave as if in anaphase.

In our view, the fact that SynSAC does not come from kinetochores is a major advantage as this allows the study of the kinetochore in an unperturbed state. It is also important to note that the canonical checkpoint components are all still present in the SynSAC strains, and perturbations in kinetochore-microtubule interactions would be expected to mount a kinetochore-driven checkpoint response as normal. Indeed, it would be interesting in future work to understand how disrupting kinetochore-microtubule attachments alters kinetochore composition (presumably checkpoint proteins will be recruited) and phosphorylation but this is beyond the scope of this work. In terms of the state at which we are arresting cells – this is a true metaphase because cohesion has not been lost but kinetochore-microtubule attachments have been established. This is evident from the enrichment of microtubule regulators but not checkpoint proteins in the kinetochore purifications from metaphase I and II. While this state is expected to occur only transiently in yeast, since the establishment of proper kinetochore-microtubule attachments triggers anaphase onset, the ability to capture this properly bioriented state will be extremely informative for future studies. We acknowledge however that we cannot completely rule out unwanted effects of the system, as in any synchronisation system, and where possible findings with the system should be backed up with an orthogonal approach. We appreciate the reviewers’ insight in highlighting these interesting discussion points and we have re-written the relevant paragraph in the discussion, starting line 545.

**Reviewer #1 (Significance):**
These authors have developed a method to induce MI or MII arrest. While this was previously possible in MI, the advantage of the method presented here is it works for MII, and chemically inducible because it is based on a system that is sensitive to the addition of ABA. Depending on when the ABA is added, they achieve a MI or MII delay. The ABA promotes dimerizing fragments of Mps1 and Spc105 that can't bind their chromosomal sites. The evidence that the MI arrest is weaker than the MII arrest is convincing and consistent with published data and indicating the SAC in MI is less robust than MII or mitosis. The authors use this system to find evidence that the weak MI arrest is associated with PP1 binding to Spc105. This is a nice use of the system.The remainder of the paper uses the SynSAC system to isolate populations enriched for MI or MII stages and conduct proteomics. This shows a powerful use of the system but more work is needed to validate these results, particularly in normal cells.Overall the most significant aspect of this paper is the technical achievement, which is validated by the other experiments. They have developed a system and generated some proteomics data that maybe useful to others when analyzing kinetochore composition at each division.

We appreciate the reviewer’s enthusiasm for our work.

**Reviewer #2 (Evidence, reproducibility and clarity):**
The manuscript submitted by Koch et al. describes a novel approach to collect budding yeast cells in metaphase I or metaphase II by synthetically activating the spinde checkpoint (SAC). The arrest is transient and reversible. This synchronization strategy will be extremely useful for studying meiosis I and meiosis II, and compare the two divisions. The authors characterized this so-named syncSACapproach and could confirm previous observations that the SAC arrest is less efficient in meiosis I than in meiosis II. They found that downregulation of the SAC response through PP1 phosphatase is stronger in meiosis I than in meiosis II. The authors then went on to purify kinetochore-associated proteins from metaphase I and II extracts for proteome and phosphoproteome analysis. Their data will be of significant interest to the cell cycle community (they compared their datasets also to kinetochores purified from cells arrested in prophase I and -with SynSAC in mitosis).I have only a couple of minor comments:(1) I would add the Suppl Figure 1A to main Figure 1A. What is really exciting here is the arrest in metaphase II, so I don't understand why the authors characterize metaphase I in the main figure, but not metaphase II. But this is only a suggestion.

Thanks for the suggestion. We agree and have moved the data for both meiosis I and meiosis II to make a new main Figure 2.

(2) Line 197, the authors state: ...SyncSACinduced a more pronounced delay in metaphase II than in metaphase I. However, line 229 and 240 the auhtors talk about a "longer delay in metaphase <i compared to metaphase II"... this seems to be a mix-up.

Thank you for pointing this out, this is indeed a typo and we have corrected it.

(3) The authors describe striking differences for both protein abundance and phosphorylation for key kinetochore associated proteins. I found one very interesting protein that seems to be very abundant and phosphorylated in metaphase I but not metaphase II, namely Sgo1. Do the authors think that Sgo1 is not required in metaphase II anymore? (Top hit in suppl Fig 8D).

This is indeed an interesting observation, which we plan to investigate as part of another study in the future. Indeed, data from mouse indicates that shugoshin-dependent cohesin deprotection is already absent in meiosis II in mouse oocytes [7], though whether this is also true in yeast is not known. Furthermore, this does not rule out other functions of Sgo1 in meiosis II (for example promoting biorientation). We have included a paragraph in the discussion in the section starting line 641.

**Reviewer #2 (Significance):**
The technique described here will be of great interest to the cell cycle community. Furthermore, the authors provide data sets on purified kinetochores of different meiotic stages and compare them to mitosis. This paper will thus be highly cited, for the technique, and also for the application of the technique.
**Reviewer #3 (Evidence, reproducibility and clarity):**
In their manuscript, Koch et al. describe a novel strategy to synchronize cells of the budding yeast *Saccharomyces cerevisiae* in metaphase I and metaphase II, thereby facilitating comparative analyses between these meiotic stages. This approach, termed SynSAC, adapts a method previously developed in fission yeast and human cells that enables the ectopic induction of a synthetic spindle assembly checkpoint (SAC) arrest by conditionally forcing the heterodimerization of two SAC components upon addition of the plant hormone abscisic acid (ABA). This is a valuable tool, which has the advantage that induces SAC-dependent inhibition of the anaphase promoting complex without perturbing kinetochores. Furthermore, since the same strategy and yeast strain can be also used to induce a metaphase arrest during mitosis, the methodology developed by Koch et al. enables comparative analyses between mitotic and meiotic cell divisions. To validate their strategy, the authors purified kinetochores from meiotic metaphase I and metaphase II, as well as from mitotic metaphase, and compared their protein composition and phosphorylation profiles. The results are presented clearly and in an organized manner.

We are grateful to the reviewer for their support.

Despite the relevance of both the methodology and the comparative analyses, several main issues should be addressed:(1) In contrast to the strong metaphase arrest induced by ABA addition in mitosis (Supp. Fig. 2), the SynSAC strategy only promotes a delay in metaphase I and metaphase II as cells progress through meiosis. This delay extends the duration of both meiotic stages, but does not markedly increase the percentage of metaphase I or II cells in the population at a given timepoint of the meiotic time course (Fig. 1C). Therefore, although SynSAC broadens the time window for sample collection, it does not substantially improve differential analyses between stages compared with a standard NDT80 prophase block synchronization experiment. Could a higher ABA concentration or repeated hormone addition improve the tightness of the meiotic metaphase arrest?

For many purposes the enrichment and extended time for sample collection is sufficient, as we demonstrate here. However, as pointed out by the reviewer below, the system can be improved by use of the 4A-RASA mutations to provide a stronger arrest (see our response below). We did not experiment with higher ABA concentrations or repeated addition since the very robust arrest achieved with the 4A-RASA mutant deemed this unnecessary.

(2) Unlike the standard SynSAC strategy, introducing mutations that prevent PP1 binding to the SynSAC construct considerably extended the duration of the meiotic metaphase arrests. In particular, mutating PP1 binding sites in both the RVxF (RASA) and the SILK (4A) motifs of the Spc105(1-455)-PYL construct caused a strong metaphase I arrest that persisted until the end of the meiotic time course (Fig. 3A). This stronger and more prolonged 4A-RASA SynSAC arrest would directly address the issue raised above. It is unclear why the authors did not emphasize more this improved system. Indeed, the 4A-RASA SynSAC approach could be presented as the optimal strategy to induce a conditional metaphase arrest in budding yeast meiosis, since it not only adapts but also improves the original methods designed for fission yeast and human cells. Along the same lines, it is surprising that the authors did not exploit the stronger arrest achieved with the 4A-RASA mutant to compare kinetochore composition at meiotic metaphase I and II.

We agree that the 4A-RASA mutant is the best tool to use for the arrest and going forward this will be our approach. We collected the proteomics data and the data on the SynSAC mutant variants concurrently, so we did not know about the improved arrest at the time the proteomics experiment was done. Because a very good arrest was already achieved with the unmutated SynSAC construct, we could not justify repeating the proteomics experiment which is a large amount of work using significant resources. We highlighted the potential of using the 4A-RASA variant more strongly as follows:

Line 312, Results:

“These findings also indicate that *spc105(1-455)-4A-RASA* is the preferred SynSAC variant, particularly where metaphase I arrest is the goal.”

Line 598, Discussion: “Finally, the stronger and more prolonged SynSAC arrest obtained using the PP1 binding site mutant *spc105(1-455)-4A-RASA* prompts its consideration as an alternative tool for future studies, particularly where meiosis I arrest is important. At the time of performing the kinetochore immunoprecipitations, these mutations were not yet available but, as we have demonstrated, wild type SynSAC protein fragments nevertheless yielded sufficiently enriched populations of metaphase I and II cells to allow reliable detection of stage-specific kinetochore proteins and phosphorylations. Going forward, however, we consider SynSAC-4A-RASA to be the optimal tool for inducing metaphase arrests.”

(3) The results shown in Supp. Fig. 4C are intriguing and merit further discussion. Mitotic growth in ABA suggest that the RASA mutation silences the SynSAC effect, yet this was not observed for the 4A or the double 4A-RASA mutants. Notably, in contrast to mitosis, the SynSAC 4A-RASA mutation leads to a more pronounced metaphase I meiotic delay (Fig. 3A). It is also noteworthy that the RVAF mutation partially restores mitotic growth in ABA. This observation supports, as previously demonstrated in human cells, that Aurora B-mediated phosphorylation of S77 within the RVSF motif is important to prevent PP1 binding to Spc105 in budding yeast as well.

We agree these are intriguing findings that highlight key differences as to the wiring of the spindle checkpoint in meiosis and mitosis and potential for future studies, however, currently we can only speculate as to the underlying cause. The effect of the RASA mutation in mitosis is unexpected and unexplained. However, the fact that the 4A-RASA mutation causes a stronger delay in meiosis I compared to mitosis can be explained by a greater prominence of PP1 phosphatase in meiosis. Indeed, our data (now Figure 7A) show that the PP1 phosphatase Glc7 and its regulatory subunit Fin1 are highly enriched on kinetochores at all meiotic stages compared to mitosis.

We agree that the improved growth of the RVAF mutant is intriguing, along with the reduced metaphase I delay, which together point to a role of Aurora B-mediated phosphorylation also in *S. cerevisiae*, though previous work has not supported such a role [8].

We have re-written and expanded the paragraph in the discussion related to the mutation of the RVSF motif starting line 564 to reflect these points.

(4) To demonstrate the applicability of the SynSAC approach, the authors immunoprecipitated the kinetochore protein Dsn1 from cells arrested at different meiotic or mitotic stages, and compared kinetochore composition using data independent acquisition (DIA) mass spectrometry. Quantification and comparative analyses of total and kinetochore protein levels were conducted in parallel for cells expressing either FLAG-tagged or untagged Dsn1 (Supp. Fig. 7A-B). To better detect potential changes, protein abundances were next scaled to Dsn1 levels in each sample (Supp. Fig. 7C-D). However, it is not clear why the authors did not normalize protein abundance in the immunoprecipitations from tagged samples at each stage to the corresponding untagged control, instead of performing a separate analysis. This would be particularly relevant given the high sensitivity of DIA mass spectrometry, which enabled quantification of thousands of proteins. Furthermore, the authors compared protein abundances in tagged-samples from mitotic metaphase and meiotic prophase, metaphase I and metaphase II (Supp. Fig. 7E-F). If protein amounts in each case were not normalized to the untagged controls, as inferred from the text (lines 333 to 338), the observed differences could simply reflect global changes in protein expression at different stages rather than specific differences in protein association to kinetochores.

While we agree with the reviewer that at first glance, normalising to no tag appears to be the most appropriate normalisation, in practice there is very low background signal in the no tag sample which means that any random fluctuations have a big impact on the final fold change used for normalisation. This approach therefore introduces artefacts into the data rather than improving normalisation.

To provide reassurance that our kinetochore immunoprecipitations are specific, and that the background (no tag) signal is indeed very low, we have provided a new figure showing the volcanos comparing kinetochore purifications at each stage with their corresponding no tag control (Figure 5).

It is also important to note that our experiment looks at relative changes of the same protein over time, which we expect to be relatively small in the whole cell lysate. We previously documented proteins that change in abundance in whole cell lysates throughout meiosis [9]. In this study, we found that relatively few proteins significantly change in abundance. We added a sentence to this effect in the discussion (Line 632). “Although some variation could reflect global changes in protein abundance during meiosis, we previously found that only a few proteins undergo dynamic abundance changes during the meiotic divisions [9], so this is unlikely to fully explain the kinetochore composition differences observed.”

Our aim in the current study was to understand how the relative composition of the kinetochore changes and for this, we believe that a direct comparison to Dsn1, a central kinetochore protein which we immunoprecipitated is the most appropriate normalisation.

(5) Despite the large amount of potentially valuable data generated, the manuscript focuses mainly on results that reinforce previously established observations (e.g., premature SAC silencing in meiosis I by PP1, changes in kinetochore composition, etc.). The discussion would benefit from a deeper analysis of novel findings that underscore the broader significance of this study.

We strongly agree with this point and we have re-framed the discussion to focus on the novel findings, as also raised by the other reviewers and noted above.

Finally, minor concerns are:(1) Meiotic progression in SynSAC strains lacking Mad1, Mad2 or Mad3 is severely affected (Fig. 1D and Supp. Fig. 1), making it difficult to assess whether, as the authors state, the metaphase delays depend on the canonical SAC cascade. In addition, as a general note, graphs displaying meiotic time courses could be improved for clarity (e.g., thinner data lines, addition of axis gridlines and external tick marks, etc.).

We added the requested data, which is now part of Figure 2. This now clearly shows that *mad2* and *mad3* mutants have very similar meiotic cell cycle profiles in the SynSAC background whether or not ABA is added. Please note that we removed the *mad1* mutant from this analysis as technical difficulties prevented the strain from entering meiosis well.

We have improved graphs throughout, as suggested: data lines are thinner, axis gridlines and external grid marks are included. We added an arrow to indicate the time of ethanol/ABA addition.

(2) Spore viability following SynSAC induction in meiosis was used as an indicator that this experimental approach does not disrupt kinetochore function and chromosome segregation. However, this is an indirect measure. Direct monitoring of genome distribution using GFP-tagged chromosomes would have provided more robust evidence. Notably, the SynSAC mad3Δ mutant shows a slight viability defect, which might reflect chromosome segregation defects that are more pronounced in the absence of a functional SAC.

Spore viability is a much more sensitive way of analysing segregation defects than GFP-labelled chromosomes. This is because GFP labelling allows only a single chromosome to be followed. On the other hand, if any of the 16 chromosomes mis-segregate in a given meiosis this would result in one or more aneuploid spores in the tetrad, which are typically inviable. The fact that spore viability is not significantly different from wild type in this analysis indicates that there are no major chromosome segregation defects in these strains, and we therefore we think this experiment unnecessary.

(3) It is surprising that, although SAC activity is proposed to be weaker in metaphase I, the levels of CPC/SAC proteins seem to be higher at this stage of meiosis than in metaphase II or mitotic metaphase (Fig. 4A-B).

We speculate that the challenge in biorienting homologs which are held together by chiasmata, rather than back-to-back kinetochores results in a greater requirement for dynamic error correction in meiosis I. Interestingly, the data with the RASA mutant also point to increased PP1 activity in meiosis I, and we additionally observed increased levels of PP1 (Glc7 and Fin1) on meiotic kinetochores, consistent with the idea that cycles of error correction and silencing are elevated in meiosis I. We have re-written and expanded the discussion section starting line 565 to reflect these points.

(4) Although a more detailed exploration of kinetochore composition or phosphorylation changes is beyond the scope of the manuscript, some key observations could have been validated experimentally (e.g., enrichment of proteins at kinetochores, phosphorylation events that were identified as specific or enriched at a certain meiotic stage, etc.).

We agree that this is beyond the scope of the current study but will form the start of future projects from our group, and hopefully others.

(5) Several typographical errors should be corrected (e.g., "Kinvetochores" in Fig. 4 legend, "250uM ABA" in Supp. Fig. 1 legend, etc.)

Thank you for pointing these out, they have been corrected and we have carefully proofread the manuscript.

**Reviewer #3 (Significance):**
Koch et al. describe a novel methodology, SynSAC, to synchronize budding yeast cells in metaphase I or metaphase II during meiosis, as well and in mitotic metaphase, thereby enabling differential analyses among these cell division stages. Their approach builds on prior strategies originally developed in fission yeast and human cells models to induce a synthetic spindle assembly checkpoint (SAC) arrest by conditionally forcing the heterodimerization of two SAC proteins upon addition of abscisic acid (ABA). The results from this manuscript are of special relevance for researchers studying meiosis and using *Saccharomyces cerevisiae* as a model. Moreover, the differential analysis of the composition and phosphorylation of kinetochores from meiotic metaphase I and metaphase II adds interest for the broader meiosis research community. Finally, regarding my expertise, I am a researcher specialized in the regulation of cell division.

References

(1) Salah, S.M., and Nasmyth, K. (2000). Destruction of the securin Pds1p occurs at the onset of anaphase during both meiotic divisions in yeast. Chromosoma *109*, 27–34.

(2) Matos, J., Lipp, J.J., Bogdanova, A., Guillot, S., Okaz, E., Junqueira, M., Shevchenko, A., and Zachariae, W. (2008). Dbf4-dependent CDC7 kinase links DNA replication to the segregation of homologous chromosomes in meiosis I. Cell *135*, 662–678.

(3) Marston, A.L.A.L., Lee, B.H.B.H., and Amon, A. (2003). The Cdc14 phosphatase and the FEAR network control meiotic spindle disassembly and chromosome segregation. Developmental cell *4*, 711–726. https://doi.org/10.1016/S1534-5807(03)00130-8.

(4) Marston, A.L., Lee, B.H., and Amon, A. (2003). The Cdc14 phosphatase and the FEAR network control meiotic spindle disassembly and chromosome segregation. Dev Cell *4*, 711–726. https://doi.org/10.1016/s1534-5807(03)00130-8.

(5) Attner, M.A., and Amon, A. (2012). Control of the mitotic exit network during meiosis. Molecular Biology of the Cell *23*, 3122–3132. https://doi.org/10.1091/mbc.E12-03-0235.

(6) Pablo-Hernando, M.E., Arnaiz-Pita, Y., Nakanishi, H., Dawson, D., del Rey, F., Neiman, A.M., and de Aldana, C.R.V. (2007). Cdc15 Is Required for Spore Morphogenesis Independently of Cdc14 in *Saccharomyces cerevisiae*. Genetics *177*, 281–293. https://doi.org/10.1534/genetics.107.076133.

(7) El Jailani, S., Cladière, D., Nikalayevich, E., Touati, S.A., Chesnokova, V., Melmed, S., Buffin, E., and Wassmann, K. (2025). Eliminating separase inhibition reveals absence of robust cohesin protection in oocyte metaphase II. EMBO J *44*, 5187–5214. https://doi.org/10.1038/s44318-025-00522-0.

(8) Rosenberg, J.S., Cross, F.R., and Funabiki, H. (2011). KNL1/Spc105 Recruits PP1 to Silence the Spindle Assembly Checkpoint. Current Biology *21*, 942–947. https://doi.org/10.1016/j.cub.2011.04.011.

(9) Koch, L.B., Spanos, C., Kelly, V., Ly, T., and Marston, A.L. (2024). Rewiring of the phosphoproteome executes two meiotic divisions in budding yeast. EMBO J *43*, 1351–1383. https://doi.org/10.1038/s44318-024-00059-8.